# Differentially private anonymized histograms

**Ananda Theertha Suresh**
Google Research, New York
theertha@google.com

## Abstract

For a dataset of label-count pairs, an anonymized histogram is the multiset of counts. Anonymized histograms appear in various potentially sensitive contexts such as password-frequency lists, degree distribution in social networks, and estimation of symmetric properties of discrete distributions. Motivated by these applications, we propose the first differentially private mechanism to release anonymized histograms that achieves near-optimal privacy utility trade-off both in terms of number of items and the privacy parameter. Further, if the underlying histogram is given in a compact format, the proposed algorithm runs in time sub-linear in the number of items. For anonymized histograms generated from unknown discrete distributions, we show that the released histogram can be directly used for estimating symmetric properties of the underlying distribution.

## 1   Introduction

Given a set of labels $\mathcal{X}$, a dataset $D$ is a collection of labels and counts, $D \stackrel{\text{def}}{=} \{(x, n_x) : x \in \mathcal{X}\}$. An anonymized histogram of such a dataset is the unordered multiset of all non-zero counts without any label information,

$$h(D) \stackrel{\text{def}}{=} \{n_x : x \in \mathcal{X} \text{ and } n_x > 0\}.$$

For example, if $\mathcal{X} = \{a, b, c, d\}$, $D = \{(a, 8), (b, 0), (c, 8), (d, 3)\}$, then $h(D) = \{3, 8, 8\}$[1]. Anonymized histograms do not contain any information about the labels, including the cardinality of $\mathcal{X}$. Furthermore, we only consider histograms with positive counts. The results can be extended to histograms that include zero counts. A histogram can also be represented succinctly using prevalences. For a histogram $h$, the *prevalence* $\varphi_r$ is the number of elements in the histogram with count $r$,

$$\varphi_r(h) \stackrel{\text{def}}{=} \sum_{n_x \in h} \mathbf{1}_{n_x = r}.$$

In the above example, $\varphi_3(h) = 1$, $\varphi_8(h) = 2$, and $\varphi_r(h) = 0$ for $r \notin \{3, 8\}$. Anonymized histograms are also referred to as histogram of histograms [1], histogram order statistics [2], profiles [3], unattributed histograms [4], fingerprints [5], and frequency lists [6].

Anonymized histograms appear in several potentially sensitive contexts ranging from password frequency lists to social networks. Before we proceed to the problem formulation and results, we first provide an overview of the various contexts where anonymized histograms have been studied under differential privacy and their motivation.

**Password frequency lists**: Anonymized password histograms are useful to security researchers who wish to understand the underlying password distribution in order to estimate the security risks or evaluate various password defenses [7, 6]. For example, if $n_{(i)}$ is the $i^{\text{th}}$ most frequent password, then $\lambda_\beta = \sum_{i=1}^{\beta} n_{(i)}$ is the number of accounts that an adversary could compromise with $\beta$ guesses

per user. Hence, if the server changes the $k$-strikes policy from 3 to 5, the frequency distribution can be used to evaluate the security implications of this change. We refer readers to [6, 8] for more uses of password frequency lists. Despite their usefulness, organizations may be wary of publishing these lists due to privacy concerns. This is further justified as it is not unreasonable to expect that an adversary will have some side information based on attacks against other organizations. Motivated by this, [6] studied the problem of releasing anonymized password histograms.

**Degree-distributions in social networks**: Degree distributions is one of the most widely studied properties of a graph as it influences the structure of the graph. Degree distribution can also be used to estimate linear queries on degree distributions such as number of $k$-stars. However, some graphs may have unique degree distributions and releasing exact degree distributions is no more safer than naive anonymization, which can leave social network participants vulnerable to a variety of attacks [9, 10, 11]. Thus releasing them exactly can be revealing. Hence, [12, 4, 13, 14, 15, 16, 17] considered the problem of releasing degree distributions of graphs with differential privacy. Degree distributions are anonymized histograms over the graph node degrees.

**Estimating symmetric properties of discrete distributions**: Let $k \stackrel{\text{def}}{=} |\mathcal{X}|$. A discrete distribution $p$ is a mapping from a domain $\mathcal{X}$ to $[0, 1]^k$ such that $\sum_x p_x = 1$. Given a discrete distribution $p$ over $k$ symbols, a *symmetric property* is a property that depends only on the multiset of probabilities [18, 19], e.g., entropy ($\sum_{x \in \mathcal{X}} p_x \log \frac{1}{p_x}$). Other symmetric properties include support size, Rényi entropy, distance to uniformity, and support coverage. Given independent samples from an unknown $p$, the goal of property estimation is to estimate the value of the symmetric property of interest for $p$. Estimating symmetric properties from unknown distributions has received a wide attention in the recent past e.g., [5, 18, 20, 21, 22, 19, 23, 24] and has applications in various fields from neuro-science [2] to genetics [25]. Recently, [26] proposed algorithms to estimate support size, support coverage and entropy with differential privacy. Optimal estimators for symmetric properties only depend on the anonymized histograms of the samples [1, 19]. Hence, releasing anonymous histograms with differential privacy would simultaneously yield differentially-private plug-in estimators for all symmetric properties.

# 2 Differential privacy

## 2.1 Definitions

Before we outline our results, we first define the privacy and utility aspects of anonymized histograms. Privacy has been studied extensively in statistics and computer science [27, 28, 29, 30] and references therein. Perhaps the most studied form of privacy is differential privacy (DP) [31, 32], where the objective is to ensure that an adversary would not infer whether a user is present in the dataset or not.

We study the problem of releasing anonymized histograms via the lens of global-DP. We begin by defining the notion of DP. Formally, given a set of datasets $\mathcal{H}$ and a notion of neighboring datasets $\mathcal{N}_{\mathcal{H}} \subseteq \mathcal{H} \times \mathcal{H}$, and a query function $f : \mathcal{H} \to \mathcal{Y}$, for some domain $\mathcal{Y}$, then a mechanism $\mathcal{M} : \mathcal{Y} \to \mathcal{O}$ is said to be $\epsilon$-DP, if for any two neighboring datasets $(h_1, h_2) \in \mathcal{N}_{\mathcal{H}}$, and all $S \subseteq \mathcal{O}$,

$$\Pr(\mathcal{M}(f(h_1)) \in S) \leq e^\epsilon \Pr(\mathcal{M}(f(h_2)) \in S). \tag{1}$$

Broadly-speaking $\epsilon$-DP ensures that given the output, an attacker would not be able to differentiate between any two neighboring datasets. $\epsilon$-DP is also called *pure*-DP and provides stricter guarantees than the approximate $(\epsilon, \delta)$-DP, where equation (1) needs to hold with probability $1 - \delta$.

Since introduction, DP has been studied extensively in various applications from dataset release to learning machine learning models [33]. It has also been adapted by industry [34]. There are two models of DP: *server* or *global* or *output* DP, where a centralized entity has access to the entire dataset and answers the queries in a DP manner. The second model is *local* DP, where $\epsilon$-DP is guaranteed for each individual user's data [35, 36, 37, 38, 39]. We study the problem of releasing anonymized histograms under global-DP. Here $\mathcal{H}$ is the set of anonymized histograms, $f$ is the identity mapping, and $\mathcal{O} = \mathcal{H}$.

## 2.2 Distance measure

For DP, a general notion of neighbors is as follows. Two datasets are neighbors if and only if one can be obtained from another by adding or removing a user [30]. Since, anonymized histograms do not

contain explicit user information, we need few definitions to apply the above notion. We first define a notion of distance between label-count datasets. A natural notion of distance between datasets $D_1$ and $D_2$ over $\mathcal{X}$ is the $\ell_1$ distance,

$$\ell_1(D_1, D_2) \overset{\text{def}}{=} \sum_{x \in \mathcal{X}} |n_x(D_1) - n_x(D_2)|,$$

where $n_x(D)$ is the count of $x$ in dataset $D$. Since anonymized histograms do not contain any information about labels, we define distance between two histograms $h_1, h_2$ as

$$\ell_1(h_1, h_2) \overset{\text{def}}{=} \min_{D_1, D_2 : h(D_1) = h_1, h(D_2) = h_2} \ell_1(D_1, D_2). \tag{2}$$

The following simple lemma characterizes the above distance in terms of counts.

**Lemma 1** (Appendix B). *For an anonymized histogram $h = \{n_x\}$, let $n_{(i)}$ be the $i^{th}$ highest count in the dataset.[2] For any two anonymized histograms $h_1, h_2$, $\ell_1(h_1, h_2) = \sum_{i \geq 1} |n_{(i)}(h_1) - n_{(i)}(h_2)|$.*

The above distance is also referred to as sorted $\ell_1$ distance or earth-mover's distance. With the above definition of distance, we can define neighbors as follows.

**Definition 1.** *Two anonymized histograms $h$ and $h'$ are neighbors if and only if $\ell_1(h, h') = 1$.*

The above definition of neighboring histograms is same as the definition of neighbors in the previous works on anonymized histograms [4, 6].

## 3 Previous and new results

### 3.1 Anonymized histogram estimation

Similar to previous works [6], we measure the utility of the algorithm in terms of the number of items in the anonymized histogram, $n \overset{\text{def}}{=} \sum_{n_x \in h} n_x = \sum_{r \geq 1} \varphi_r(h) r$.

**Previous results**: The problem of releasing anonymized histograms was first studied by [12, 4] in the context of degree distributions of graphs. They showed that adding Laplace noise to each count, followed by a post-processing isotonic regression step results in a histogram $H$ with expected sorted-$\ell_2^2$ error of

$$\mathbb{E}[\ell_2^2(h, H)] = \mathbb{E}\left[\sum_{i \geq 1} (n_{(i)}(h_1) - n_{(i)}(h_2))^2\right] = \sum_{r \geq 0} \mathcal{O}\left(\frac{\log^3 \max(\varphi_r, 1)}{\epsilon^2}\right) = \mathcal{O}\left(\frac{\sqrt{n}}{\epsilon^2}\right).$$

Their algorithm runs in time $\mathcal{O}(n)$. The problem was also considered in the context of password frequency lists by [6]. They observed that an exponential mechanism over integer partitions yields an $\epsilon$-DP algorithm. Based on this, for $\epsilon = \Omega(1/\sqrt{n})$, they proposed a dynamic programming based relaxation of the exponential mechanism that runs in time $\mathcal{O}\left(\frac{n^{3/2}}{\epsilon} + n \log \frac{1}{\delta}\right)$ and returns a histogram $H$ such that $\ell_1(h, H) = \mathcal{O}\left(\frac{\sqrt{n} + \log \frac{1}{\delta}}{\epsilon}\right)$, with probability $\geq 1 - \delta$. Furthermore, the relaxed mechanism is $(\epsilon, \delta)$-DP.

The best information-theoretic lower bound for the $\ell_1$ utility of any $\epsilon$-DP mechanism is due to [40], who showed that for $\epsilon \geq \Omega(1/n)$, any $\epsilon$-DP mechanism has expected $\ell_1$ error of $\Omega(\sqrt{n}/\sqrt{\epsilon})$ for some dataset.

**New results**: Following [6], we study the problem in $\ell_1$ metric. We propose a new DP mechanism PRIVHIST that satisfies the following:

**Theorem 1.** *Given a histogram in the prevalence form $h = \{(r, \varphi_r) : \varphi_r > 0\}$, PRIVHIST returns a histogram $H$ and a sum count $N$ that is $\epsilon$-DP. Furthermore, if $\epsilon > 1$, then*

$$\mathbb{E}[\ell_1(h, H)] = \mathcal{O}\left(\sqrt{n} \cdot e^{-c\epsilon}\right) \quad and \quad \mathbb{E}[|N - n|] \leq e^{-c\epsilon}$$

*for some constant $c > 0$ and has an expected run time of $\tilde{\mathcal{O}}(\sqrt{n})$. If $1 \geq \epsilon = \Omega(1/n)$ then,*

$$\mathbb{E}[\ell_1(h, H)] = \mathcal{O}\left(\sqrt{\frac{n}{\epsilon} \log \frac{2}{\epsilon}}\right) \quad and \quad \mathbb{E}[|N - n|] \leq \mathcal{O}\left(\frac{1}{\epsilon}\right),$$

*and has an expected run time of $\tilde{\mathcal{O}}\left(\sqrt{\frac{n}{\epsilon}} + \frac{1}{\epsilon}\right)$.*

Together with the lower bound of [40], this settles the optimal privacy utility trade-off for $\epsilon \in [\Omega(1/n), 1]$ up to a multiplicative factor of $\mathcal{O}(\sqrt{\log(2/\epsilon)})$. We also show that PRIVHIST is near-optimal for $\epsilon > 1$, by showing the following lower bound.

**Theorem 2** (Appendix E). *For a given $n$, let $\mathcal{H} = \{h : n \leq \sum_r r\varphi_r(h) \leq n + 1\}$. For any $\epsilon$-DP mechanism $\mathcal{M}$, there exists a histogram $h \in \mathcal{H}$, such that*

$$\mathbb{E}[\ell_1(h, \mathcal{M}(h))] \geq \Omega(\sqrt{n}e^{-2\epsilon}).$$

Theorems 1 and 2 together with [40] show that the the proposed mechanism has near-optimal utility for all $\epsilon = \Omega(1/n)$. We can infer the number of items in the dataset by $\sum_r r \cdot \varphi_r(H)$. However, this estimate is very noisy. Hence, we also return the sum of counts $N$ as it is useful for applications in symmetric property estimation for distributions. Apart from the near-optimal privacy-utility trade-off, we also show that PRIVHIST has several other useful properties.

**Time complexity**: By the Hardy-Ramanujan integer partition theorem [41], the number of anonymized histograms with $n$ items is $e^{\Theta(\sqrt{n})}$. Hence, we can succinctly represent them using $\mathcal{O}(\sqrt{n})$ space. Recall that any anonymized histogram can be written as $\{(r, \varphi_r) : \varphi_r > 0\}$, where $\varphi_r$ is the number of symbols with count $r$. Let $t$ be the number of distinct counts and let $r_1, r_2, \ldots, r_t$ be the distinct counts with non-zero prevalences. Then $r_i \geq i$ and

$$n = \sum_{i=1}^{t} r_i \varphi_{r_i} \geq \sum_{i=1}^{t} r_i \geq \sum_{i=1}^{t} i \geq \frac{t^2}{2},$$

and hence there are at most $t \leq \sqrt{2n}$ non-zero prevalences and $h$ can be represented as $\{(r, \varphi_r) : \varphi_r > 0\}$ using $\mathcal{O}(\sqrt{n})$ count-prevalence pairs. Histograms are often stored in this format for space efficiency e.g., password frequency lists in [42]. PRIVHIST takes advantage of this succinct representation. Hence, given such a succinct representation, it runs time $\mathcal{O}(\sqrt{n})$ as opposed to the $\mathcal{O}(n)$ running time of [12] and $\mathcal{O}(n^{3/2})$ running time of [6]. This is highly advantageous for large datasets such as password frequency lists with $n = 70M$ data points [6].

**Pure vs approximate differential privacy**: The only previous known algorithm with $\ell_1$ utility of $\mathcal{O}(\sqrt{n})$ is that of [6] and it runs in time $\mathcal{O}(n^{3/2})$. However, their algorithm is $(\epsilon, \delta)$-approximate DP which is strictly weaker than PRIVHIST, whose output is $\epsilon$-DP. For applications in social networks it is desirable to have group privacy for large groups [32]. For groups of size $k$, $(\epsilon, \delta)$ approximate DP, scales as $(k\epsilon, e^{k\epsilon}\delta)$-DP, which can be prohibitive for large values of $k$. Hence $\epsilon$-DP is preferable.

**Applications to symmetric property estimation**: We show that the output of PRIVHIST can be directly applied to obtain near-optimal sample complexity algorithms for discrete distribution symmetric property estimation.

### 3.2 Symmetric property estimation of discrete distributions

For a symmetric property $f$ and an estimator $\hat{f}$ that uses $n$ samples, let $\mathcal{E}(\hat{f}, n)$ be an upper bound on the worst expected error over all distributions $p$ with support at most $k$, $\mathcal{E}(\hat{f}, n) \stackrel{\text{def}}{=} \max_{p \in \Delta^k} \mathbb{E}[|f(p) - \hat{f}(X^n)|]$ . Let sample complexity $n(f, \alpha)$ denote the minimum number of samples such that $\mathcal{E}(\hat{f}, n) \leq \alpha$, $n(f, \alpha) \stackrel{\text{def}}{=} \min\{n : \mathcal{E}(\hat{f}, n) \leq \alpha\}$.

Given samples $X^n \stackrel{\text{def}}{=} X_1, X_2, \ldots, X_n$, let $h(X^n)$ denote the corresponding anonymous histogram. For a symmetric property $f$, linear estimators of the form

$$\hat{f}(h(X)) \stackrel{\text{def}}{=} \sum_{r \geq 1} f(r, n) \cdot \varphi_r(h(X^n),$$

are shown to be sample-optimal for symmetric properties such as entropy [21], support size [18, 20], support coverage [22], and Rényi entropy [43, 44], where $f(r, n)$s are some distribution-independent coefficients that depend on the property $f$. Recently, [26] showed that for any given property such as entropy or support size, one can construct DP estimators by adding Laplace noise to the non-private estimator. They further showed that this approach is information theoretically near-optimal.

Instead of just computing a DP estimate for a given property, the output of PRIVHIST can be directly used to estimate any symmetric property. By the post-processing lemma [32], since the output of PRIVHIST is DP, the estimate is also DP. For an estimator $\hat{f}$, let $L_{\hat{f}}^n$ be the Lipschitz constant given by $L_{\hat{f}}^n \overset{\text{def}}{=} \max(f(1, n), \max_{r \geq 1} |f(r, n) - f(r + 1, n)|)$. If instead of $h(X^n)$, a DP histogram $H$ and the sum of counts $N$ is available, then $\hat{f}$ can be modified as

$$\hat{f}^{dp} \overset{\text{def}}{=} \sum_{r \geq 1} f(r, N) \cdot \varphi_r(H),$$

which is differentially private. Using Theorem 1, we show that:

**Corollary 1** (Appendix F). *Let $\hat{f}$ satisfy $L_{\hat{f}}^n \leq n^{\beta-1}$, for a $\beta \leq 0.5$. Further, let there exists $\mathcal{E}$ such that $|\mathcal{E}(\hat{f}, n) - \mathcal{E}(\hat{f}, n + 1)| \leq n^{\beta-1}$. Let $f_{\max} = \max_{p \in \Delta^k} f(p)$. If $n(\hat{f}, \alpha)$ is the sample complexity of estimator $\hat{f}$, then for $\epsilon > 1$*

$$n(\hat{f}^{dp}, 2\alpha) \leq \max\left( n(\hat{f}, \alpha), \mathcal{O}\left( \left(\frac{1}{\alpha e^{c\epsilon}}\right)^{\frac{2}{1-2\beta}} + \frac{1}{\epsilon} \log \frac{f_{\max}}{\alpha} \right) \right).$$

*for some constant $c > 0$. For $\Omega(1/n) \leq \epsilon \leq 1$,*

$$n(\hat{f}^{dp}, 2\alpha) \leq \max\left( n(\hat{f}, \alpha), \mathcal{O}\left( \left(\frac{\log(2/\epsilon)}{\alpha^2 \epsilon}\right)^{\frac{1}{1-2\beta}} + \frac{1}{\epsilon} \log \frac{f_{\max}}{\alpha\epsilon} \right) \right).$$

*Further, by the post-processing lemma, $\hat{f}^{dp}$ is also $\epsilon$-DP.*

For entropy $(-\sum_x p_x \log p_x)$, normalized support size $(\sum_x \mathbf{1}_{p_x > 1/k}/k)$, and normalized support coverage, there exists sample-optimal linear estimators with $\beta < 0.1$ and have the property $|\mathcal{E}(\hat{f}, n) - \mathcal{E}(\hat{f}, n + 1)| \leq \mathcal{E}(\hat{f}, n)n^{\beta-1}$ [19, 26]. Hence the sample complexity of the proposed algorithm increases at most by a polynomial in $1/\epsilon\alpha$. Furthermore, the increase is dependent on the maximum value of the function for distributions of interest and it does not explicitly depend on the support size. This result is slightly worse than the property specific results of [26] in terms of dependence on $\epsilon$ and $\alpha$. In particular, for entropy estimation, the main term in our privacy cost is $\tilde{\mathcal{O}}\left( (1/\alpha^2\epsilon)^{\frac{1}{1-2\beta}} \right)$ and the bound of [26] is $\mathcal{O}\left( 1/(\alpha\epsilon)^{1+\beta} \right)$. Thus for $\beta = 0.1$, our dependence on $\epsilon$ and $\alpha$ is slightly worse. However, we note that our results are more general in that $H$ can be used with any linear estimator. For example, our algorithm implies DP algorithms for estimating distance to uniformity, which have been not been studied before. Furthermore, PRIVHIST can also be combined with the maximum likelihood estimators of [3, 22, 45] and linear programming estimators of [5], however we do not provide any theoretical guarantees for these combined algorithms.

## 4  PRIVHIST

In the algorithm description and analysis, let $\bar{x}$ denote the vector $x$ and let $\varphi_{r+}(h) \overset{\text{def}}{=} \sum_{s \geq r} \varphi_s(h)$ denote the cumulative prevalences. Since, anonymized histograms are multisets, we can define the sum of two anonymized histograms as follows: for two histograms $h_1, h_2$, the sum $h = h_1 + h_2$ is given by $\varphi_r(h) = \varphi_r(h_1) + \varphi_r(h_2), \forall r$. Furthermore, since there is a one-to-one mapping between histograms in count form $h = \{n_{(i)}\}$ and in prevalence form $h = \{(r, \varphi_r) : \varphi_r > 0\}$, we use both interchangeably. For the ease of analysis, we also use the notation of improper histogram, where the $\varphi_r$'s can be negative or non-integers. Finally, for a histogram $h^a$ indexed by super-script $a$, we define $\varphi^a \overset{\text{def}}{=} \varphi(h^a)$ for the ease of notation.

## 4.1 Approach

Instead of describing the technicalities involved in the algorithm directly, we first motivate the algorithm with few incorrect or high-error algorithms. Before we proceed, recall that histograms can be written either in terms of prevalences $\varphi_r$ or in terms of sorted counts $n_{(i)}$.

**An incorrect algorithm**: A naive tendency would be to just add noise only to the non-zero prevalences or counts. However, this is not differentially private. For example, consider two neighboring histograms in prevalence format, $h = \{\varphi_1 = 2\}$ and $h' = \{\varphi_1 = 1, \varphi_2 = 1\}$. The resulting outputs for the above two inputs would be very different as the output of $h$ never produces a non-zero $\varphi_2$, whereas the output of $h'$ produces non-zero $\varphi_2$ with high probability. Similar counter examples can be shown for sorted counts.

**A high-error algorithm**: Instead of adding noise to non-zero counts or prevalences, one can add noise to all the counts or prevalences. It can be shown that adding noise to all the counts (including those appeared zero times), yields a $\ell_1$ error $\mathcal{O}(n/\epsilon)$, whereas adding noise to prevalences can yield an $\ell_1$ error of $\mathcal{O}(n^2/\epsilon)$, if we naively use the utility bound in terms of prevalences (3). We note that [12] showed that a post-processing step after adding noise to sorted-counts and improves the $\ell_2$ utility. A naive application of the Cauchy-Schwarz inequality yields an $\ell_1$ error of $n^{3/4}/\epsilon$ for that algorithm. While it might be possible to improve the dependence on $n$ by a tighter analysis, it is not clear if the dependence on $\epsilon$ can be improved.

The algorithm is given in PRIVHIST. After some computation, it calls two sub-routines PRIVHIST-LOWPRIVACY and PRIVHIST-HIGHPRIVACY depending on the value of $\epsilon$. PRIVHIST has two main new ideas: (i) splitting $r$ around $\sqrt{n}$ and using prevalences in one regime and counts in another and (ii) the smoothing technique used to zero out the prevalence vector. Of the two (i) is crucial for the computational complexity of the algorithm and (ii) is crucial in improving the $\epsilon$-dependence from $1/\epsilon$ to $1/\sqrt{\epsilon}$ in the high privacy regime ($\epsilon \leq 1$). There are more subtle differences such as using cumulative prevalences instead of actual prevalences. We highlight them in the next section. We now overview our algorithm for low and high privacy regimes separately.

## 4.2 Low privacy regime

We first consider the problem in the low privacy regime when $\epsilon > 1$. We make few observations.

**Geometric mechanism vs Laplace mechanism**: For obtaining DP output of integer data, one can add either Laplace noise or geometric noise [46]. For $\epsilon$-DP, the expected $\ell_1$ noise added by Laplace mechanism is $1/\epsilon$, which strictly larger than that of the geometric mechanism $(2e^{-\epsilon}/(1 - e^{-2\epsilon}))$ (see Appendix A). For $\epsilon > 1$, we use the geometric mechanism to obtain optimal trade off in terms of $\epsilon$.

**Prevalences vs counts**: If we add noise to each coordinate of a $d$-dimensional vector, the total amount of noise in $\ell_1$ norm scales linearly in $d$, hence it is better to add noise to a small dimensional vector. In the worst case, both prevalences and counts can be an $n$-dimensional vector. Hence, we propose to use prevalences for small values of $r \leq \sqrt{n}$, and use counts for $r > \sqrt{n}$. This ensures that the dimensionality of vectors to which we add noise is at most $\mathcal{O}(\sqrt{n})$.

**Cumulative prevalences vs prevalences**: The $\ell_1$ error can be bounded in terms of prevalences as follows. See Appendix B for a proof.

$$\ell_1(h_1, h_2) \leq \sum_{r \geq 1} |\varphi_{r+}(h_1) - \varphi_{r+}(h_2)| \leq \sum_{r \geq 1} r |\varphi_r(h_1) - \varphi_r(h_2)|, \tag{3}$$

If we add noise to prevalences, the $\ell_1$ error can be very high as noise is multiplied with the corresponding count $r$ (3) . The bound in terms of cumulative prevalences $\varphi_+$ is much tighter. Hence, for small values of $r$, we use cumulative prevalences instead of prevalences themselves.

The above observations provide an algorithm for the low privacy regime. However, there are few technical difficulties. For example, if we split the counts at a threshold naively, then it is not differentially private. We now describe each of the steps in the high-privacy regime.

**(1) Find $\sqrt{n}$**: To divide the histogram into two smaller histograms, we need to know $n$, which may not be available. Hence, we allot $\epsilon_1$ privacy cost to find a DP value of $n$.

**(2) Sensitivity preserving histogram split**: If we divide the histogram into two parts based on counts naively and analyze the privacy costs independently for the higher and smaller parts separately,

then the sensitivity would be lot higher for certain neighboring histograms. For example, consider two neighboring histograms $h_1 = \{\varphi_T = 1, \varphi_{n-T} = 1\}$ and $h_2 = \{\varphi_{T+1} = 1, \varphi_{n-T-1} = 1\}$. If we divide $h_1$ in to two parts based on threshold $T$, say $h_1^s = \{\varphi_T = 1\}$ and $h_1^\ell = \{\varphi_{n-T} = 1\}$ and $h_2^s = \{\}$ and $h_2^\ell = \{\varphi_{T+1} = 1, \varphi_{n-T-1} = 1\}$, then $\ell_1(h_1^\ell, h_2^\ell) = T + 2$. Thus, the $\ell_1$ distance between neighboring separated histograms $\ell_1(h_1^\ell, h_2^\ell), \ell_1(h_1^s, h_2^s)$ would be much higher compared to $\ell_1(h_2, h_2)$ and we need to add a lot of noise. Therefore, we perturb $\varphi_T$ and $\varphi_{T+1}$ using geometric noise. This ensures DP in instances where the neighboring histograms differ at $\varphi_T$ and $\varphi_{T+1}$, and doesn't change the privacy analysis for other types of histograms. However, adding noise may make the histogram improper as $\varphi_T$ can become negative. To this end, we add $M$ fake counts at $T$ and $T + 1$ to ensure that the histogram is proper with high probability. We remove them later in **(L4)**. We refer readers to Appendix C.2 for details about this step.

**(3,4) Add noise**: Let $H^{bs}$ (small counts) and $H^{b\ell}$ (large counts) be the split-histograms. We add noise to cumulative prevalences in $H^{bs}$ and counts in $H^{b\ell}$ as described in the algorithm overview.

**(L1, L2) Post-processing**: The noisy versions of $\varphi_{r+}$ may not satisfy the properties satisfied by the histograms i.e., $\varphi_{r+} \geq \varphi_{(r+1)+}$. To overcome this, we run isotonic regression over noisy $\varphi_{r+}$ subject to the monotonicity constraints i.e., given noisy counts $\varphi_{r+}$, find $\varphi_{r+}^{\text{mon}}$ that minimizes $\sum_{r \leq T}(\varphi_{r+} - \varphi_{r+}^{\text{mon}})^2$, subject to the constraint that $\varphi_{r+}^{\text{mon}} \geq \varphi_{(r+1)+}^{\text{mon}}$, for all $r \leq T$. Isotonic regression in one dimension can be run in time linear in the number of inputs using the pool adjacent violators algorithm (PAVA) or its variants [47, 48]. Hence, the time complexity of this algorithm is $\mathcal{O}(T) \approx \sqrt{n}$. We then round the prevalences to the nearest non-negative integers. We similarly post-process large counts and remove the fake counts that we introduced in step **(2)**.

Since we used succinct representation of histograms, used prevalences for $r$ smaller than $\mathcal{O}(\sqrt{n})$ and counts otherwise, the expected run time of the algorithm is $\tilde{\mathcal{O}}(\sqrt{n})$ for $\epsilon > 1$.

## 4.3 High privacy regime

For the high-privacy regime, when $\epsilon \leq 1$, all known previous algorithms achieve an error of $1/\epsilon$. To reduce the error from $1/\epsilon$ to $1/\sqrt{\epsilon}$, we use smoothing techniques to reduce the sensitivity and hence reduce the amount of added noise.

**Smoothing method**: Recall that the amount of noise added to a vector depends on its dimensionality. Since prevalences have length $n$, the amount of $\ell_1$ noise would be $\mathcal{O}(n/\epsilon)$. To improve on this, we first smooth the input prevalence vector such that it is non-zero only for few values of $r$ and show that the smoothing method also reduces the sensitivity of cumulative prevalences and hence reduces the amount of noise added.

While applying smoothing is the core idea, two questions remain: how to select the location of non-zero values and how to smooth to reduce the sensitivity? We now describe these technical details.

**(H1) Approximate high prevalences**: Recall that $N$ was obtained by adding geometric noise to $n$. In the rare case that this geometric noise is very negative, then there can be prevalences much larger than $2N$. This can affect the smoothing step. To overcome this, we move all counts above $2N$ to $N$. Since this changes the histogram with low probability, it does not affect the $\ell_1$ error.

**(H2) Compute boundaries**: We find a set of boundaries $S$ and smooth counts to elements in $S$. Ideally we would like to ensure that there is a boundary close to every count. For small values of $r$, we ensure this by adding all the counts and hence there is no smoothing. If $r \approx \sqrt{n}$, we use boundaries that are uniform in the log-count space. However, using this technique for all values of $r$, results in an additional $\log n$ factor. To overcome this, for $r \gg \sqrt{n}$, we use the noisy large counts in step **(4)** to find the boundaries and ensure that there is a boundary close to every count.

**(H3) Smooth prevalences**: The main ingredient in proving better utility in the high privacy regime is the smoothing technique, which we describe now with an example. Suppose that all histograms have non-zero prevalences only between counts $s$ and $s + t$ and further suppose we have two neighboring histograms $h^1$ and $h^2$ as follows. $\varphi_r^1 = 1$ and for all $i \in [s, s + t] \setminus \{r\}$, $\varphi_i^1 = 0$. Similarly, let $\varphi_{r+1}^2 = 1$ and for all $i \in [s, s + t] \setminus \{r + 1\}$, $\varphi_i^2 = 0$. If we want to release the prevalences or cumulative prevalences, we add $\ell_1$ noise of $\mathcal{O}(1/\epsilon)$ for each prevalence in $[s, s + t]$. Thus the $\ell_1$ norm of the noise would be $\mathcal{O}(t/\epsilon)$. We propose to reduce this noise by smoothing prevalences.

For a $r \in [s, s+t]$, we divide $\varphi_r$ into $\varphi_s$ and $\varphi_{s+t}$ as follows. We assign $\frac{s+t-r}{t}$ fraction of $\varphi_r$ to $\varphi_s$ and the remaining to $\varphi_{s+t}$. After this transformation, the first histogram becomes $h^{t1}$ given by, $\varphi_s^{t1} = \frac{t+s-r}{t}$ and $\varphi_s^{t1} = \frac{r}{t}$ and all other prevalences are zero. Similarly, the second histogram becomes $h^{t2}$ given by, $\varphi_s^{t2} = \frac{t+s-r-1}{t}$ and $\varphi_s^{t1} = \frac{r+1}{t}$ and all other prevalences are zero. Thus the prevalences after smoothing differ only in two locations $s$ and $s+t$ and they differ by at most $1/t$. Thus the total amount of noise needed for a DP release is $\mathcal{O}(1/t\epsilon)$ to these two prevalences. However, note that we also incur a loss as due to smoothing, which can be shown to be $\mathcal{O}(t)$. Hence, the total amount of error would be $\mathcal{O}(1/(t\epsilon) + t)$. Choosing $t = 1/\sqrt{\epsilon}$, yields a total error of $\mathcal{O}(1/\sqrt{\epsilon})$. The above analysis is for a toy example and extending it to general histograms requires additional work. In particular, we need to find the smoothing boundaries that give the best utility. As described above, we choose boundaries based on logarithmic partition of counts and also by private values of counts. The utility analysis with these choice of boundaries is in Appendix D.2.

**(H4) Add small noise**: Since the prevalences are smoothed, we add small amount of noise to the corresponding cumulative prevalences. For $\varphi_{s_i+}$, we add $L(1/(s_i - s_{i-1})\epsilon)$ to obtain $\epsilon$-DP.

**(H5) Post-processing**: Finally, we post-process the prevalences similar to **(L1)** to impose monotonicity and ensure that the resulting prevalences are positive and non-negative integers.

Since we used succinct histogram representation, ensured that the size of $S$ is small, and used counts larger than $\tilde{\mathcal{O}}(\sqrt{n\epsilon})$ to find boundaries, the expected run time is $\tilde{\mathcal{O}}\big(\sqrt{\frac{n}{\epsilon}} + \frac{1}{\epsilon}\big)$ for $\epsilon \leq 1$.

**Privacy budget allocation:** We allocate $\epsilon_1$ privacy budget to estimate $n$, $\epsilon_2$ to the rest of PRIVHIST and $\epsilon_3$ to PRIVHIST-HIGHPRIVACY. We set $\epsilon_1 = \epsilon_2 = \epsilon_3$ in our algorithms. We note that there is no particular reason for $\epsilon_1$, $\epsilon_2$, and $\epsilon_3$ to be equal and we chose those values for simplicity and easy readability. For example, since $\epsilon_1$ is just used to estimate $n$, the analysis of the algorithm shows that $\epsilon_2, \epsilon_3$ affects utility more than $\epsilon_1$. Hence, we can set $\epsilon_2 = \epsilon_3 = \epsilon(1 - o(1))/2$ and $\epsilon_1 = o(\epsilon)$ to get better practical results. Furthermore, for low privacy regime, the algorithm only uses a privacy budget of $\epsilon_1 + \epsilon_2$. Hence, we can set $\epsilon_1 = o(\epsilon)$, $\epsilon_2 = \epsilon(1 - o(1))$, and $\epsilon_3 = 0$.

## 5  Acknowledgments

Authors thank Jayadev Acharya and Alex Kulesza for helpful comments and discussions.

---

Algorithm PRIVHIST

**Input**: anonymized histogram $h$ in terms of prevalences i.e., $\{(r, \varphi_r) : \varphi_r > 0\}$, privacy cost $\epsilon$.
**Parameters**: $\epsilon_1 = \epsilon_2 = \epsilon_3 = \epsilon/3$.
**Output**: DP anonymized histogram $H$ and $N$ (an estimate of $n$).

1. DP value of the total sum: $N = \max(\sum_{n_x \in h} n_x + Z^a, 0)$, where $Z^a \sim G(e^{-\epsilon_1})$. If $N = 0$, output empty histogram and $N$. Otherwise continue.

2. Split $h$: Let $T = \lceil \sqrt{N \min(\epsilon, 1)} \rceil$ and $M = \left\lceil \frac{\max(2 \log N e^{\epsilon_2}, 1)}{\epsilon_2} \right\rceil$.

   (a) $H^a : \varphi_T^a = \varphi_T + M, \varphi_{T+1}^a = \varphi_{T+1} + M$ and $\forall r \notin \{T, T+1\}, \varphi_r^a = \varphi_r$.
   (b) $H^b : \varphi_{T+1}^b = \varphi_{T+1}^a + Z^b, \varphi_T^b = \varphi_T^a - Z^b$ and $\forall r \notin \{T, T+1\} \; \varphi_r^b = \varphi_r^a$, where $Z^b \sim G(e^{-\epsilon_2})$.
   (c) Divide $H^b$ into two histograms $H^{bs}$ and $H^{b\ell}$. For all $r \geq T+1$, $\varphi_r^{b\ell} = \max(0, \sum_{s=T+1}^r \varphi_r^b - \sum_{s=T+1}^{r-1} \varphi_r^{b\ell})$ for all $r \leq T \; \varphi_r^{bs} = \max(0, \sum_{s=r}^T \varphi_r^b - \sum_{s=r+1}^T \varphi_r^{bs})$.

3. DP value of $H^{bs}$. Let $Z_r^{cs} \sim G(e^{-\epsilon_2})$ i.i.d. and $H^{cs}$ be $\varphi_{r+}^{cs} = \varphi_{r+}^{bs} + Z_r^{cs}$ for $r \leq T$.

4. DP value of $H^{b\ell}$: Let $Z_i^{c\ell} \sim G(e^{-\epsilon_2})$ i.i.d. and $H^{c\ell}$ be $N_i^{c\ell} = N_{(i)}^{b\ell} + Z_i^{c\ell}$ for $N_{(i)} \in H^{b\ell}$.

5. If $\epsilon > 1$, output PRIVHIST-LOWPRIVACY$(H^{cs}, H^{c\ell}, T, M)$ and $N$.

6. If $\epsilon \leq 1$, output PRIVHIST-HIGHPRIVACY$(h, H^{c\ell}, T, N, \epsilon_3)$ and $N$.

<div style="border:1px solid">

Algorithm PRIVHIST-LOWPRIVACY

**Input**: low-count histogram $H^{cs}$, high-count histogram $H^{c\ell}, T, M$ and **Output**: a histogram $H$.

L1. Post processing of $H^{cs}$:

  (a) Find $\bar{\varphi}^{\mathrm{mon}}$ that minimizes $\sum_{r \geq 1}(\varphi_{r+}^{\mathrm{mon}} - \varphi_{r+}(H^{cs}))^2$. with $\varphi_{r+}^{\mathrm{mon}} \geq \varphi_{(r+1)+}^{\mathrm{mon}}, \forall r$.

  (b) $H^{ds}$: for all $r$, $\varphi_{r+}^{ds} = \mathrm{round}(\max(\varphi_{r+}^{\mathrm{mon}}, 0))$.

L2. Post processing of $H^{c\ell}$: Compute $H^{d\ell} = \{\max(N_i(H^{c\ell}), T), \forall i\}$.

L3. Let $H^d = H^{ds} + H^{d\ell}$.

L4. Compute $H^e$ by removing $M$ elements closest to $T+1$ from $H^d$ and then removing $M$ elements closest to $T$ and output it.

</div>

<div style="border:1px solid">

Algorithm PRIVHIST-HIGHPRIVACY

**Input**: non-private histogram $h$, high-count histogram $H^{\ell}, T, N, \epsilon_3$ and **Output**: a histogram $H$.

H1. Approximate higher prevalences: for $r < 2N$, $\varphi_r^u = \varphi_r(h)$ and $\varphi_{2N}^u = \varphi_{2N+}(h)$.

H2. Compute boundaries: Let the set $S$ be defined as follows:

  (a) $T' = \lceil 10\sqrt{N/\epsilon_3^3}\rceil$, $q = \sqrt{\log(1/\epsilon_3)/N\epsilon_3}$

  (b) $S = \{1, 2, \ldots, T\} \cup \{\lfloor T(1+q)^i \rfloor : i \leq \log_{1+q}(T'/T)\} \cup \{N_x : N_x \in H^{\ell}, N_x \geq T'\} \cup \{2N\}$.

H3. Smooth prevalences: Let $s_i$ denote the $i^{\mathrm{th}}$ smallest element in $S$.

  (a) $\varphi_{s_i}^v = \sum_{j=s_i}^{s_{i+1}} \varphi_j^u \cdot \frac{s_{i+1}-j}{s_{i+1}-s_i} + \sum_{j=s_{i-1}}^{s_i-1} \varphi_j^u \cdot \frac{j-s_{i-1}}{s_i-s_{i-1}}$ and if $j \notin S$, $\varphi_j^v = 0$.

H4. DP value of $H^v$: for each $s_i \in S$, let $\varphi_{s_i+}^w = \varphi_{s_i+}^v + Z^{s_i}$, where $Z^{s_i} \sim L\left(\frac{1}{\epsilon_3(s_i-s_{i-1})}\right)$.

H5. Find $H^x$ that minimizes $\sum_{s_i \in S}(\varphi_{s_i+}^x - \varphi_{s_i+}^w)^2(s_i-s_{i-1})^2$ such that $\varphi_{s_i+}^x \geq \varphi_{s_{i+1}+}^x \forall i$.

H6. Return $H^y$ given by, $\varphi_{r+}^y = \mathrm{round}(\max(\varphi_{r+}^x, 0)) \; \forall r$.

</div>

## Footnotes

[1] $h(D)$ is a multiset and not a set and duplicates are allowed.

[2]For $i$ larger than number of counts in $h$, $n_{(i)} = 0$.

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
