[Supplementary Material · supp_dphist.pdf]

# Appendix: Differentially private anonymized histograms

## A  Geometric mechanism

The mostly popular mechanism for $\epsilon$-DP is the Laplace mechanism, which is defined as follows.

**Definition 2** (Laplace mechanism ($L(b)$) [32])**.** *When the true query result is $f$, the mechanism outputs $f + Z$ where $Z$ is a random variable distributed as a Laplace distribution distribution:* $\Pr(Z = z) = \frac{1}{2b} \exp\left(-\frac{|x|}{b}\right)$ *for every $z \in \mathbb{R}$. If output of $f$ has sensitivity $\Delta$, then to achieve $\epsilon$-DP add $Z \sim L(\Delta/\epsilon)$.*

Since, we have integer inputs, we use the geometric mechanism:

**Definition 3** (Geometric mechanism ($G(\alpha)$) [46])**.** *When the true query result is $f$, the mechanism outputs $f + Z$ where $Z$ is a random variable distributed as a two-sided geometric distribution:* $\Pr(Z = z) = \frac{1-\alpha}{1+\alpha} \cdot \alpha^{|z|}$ *for every integer $z$. If output of $f$ is integers and has sensitivity $\Delta$ (an integer), then to achieve $\epsilon$-DP add $Z \sim G(e^{\epsilon/\Delta})$.*

[46] showed that geometric mechanism is universally optimal for a general class of functions under a Bayesian framework. Geometric mechanism is beneficial over Laplace mechanism in two ways: the output space of the mechanism is discrete. Since we have integer inputs, this removes the necessity of adding rounding off the outputs. For $\epsilon$-DP, the expected $\ell_1$ noise added by the Laplace mechanism is $1/\epsilon$, which strictly larger than that of the geometric mechanism ($2e^{-\epsilon}/(1 - e^{-2\epsilon})$) (see below). For moderate values of $\epsilon$, this difference is a constant. We now state few properties of the geometric distribution which are used in the rest of the paper.

We find the following set of equations useful in the rest of the paper. In the following let $Z_G \sim G(e^{-\epsilon})$ be a geometric random variable and $Z_L \sim L(1/\epsilon)$ be a Laplace random variable.

$$\mathbb{E}[Z_G] = 0 = \mathbb{E}[Z_L].$$

$$\mathbb{E}[|Z_G|] = \frac{2e^{-\epsilon}}{1 - e^{-2\epsilon}} \leq \frac{1}{\epsilon} = \mathbb{E}[|Z_L|].$$

$$\mathbb{E}[Z_G^2] = \frac{2e^{-\epsilon}}{(1 - e^{-\epsilon})^2} \leq \frac{2}{\epsilon^2} = \mathbb{E}[Z_L^2].$$

The next lemma bounds moments of $\max(n + Z, 0)$ when $Z$ is a zero mean random variable.

**Lemma 2.** *Let $Z$ be a random variable and $n \geq 0$. If $Y = \max(n + Z, 0)$, then*

$$\mathbb{E}[|Y - n|] \leq \mathbb{E}|Z|,$$

*and*

$$\mathbb{E}\left[\frac{\mathbf{1}_{Y>0}}{Y}\right] \leq \frac{1}{n} + \frac{\mathbb{E}[Z^2]}{2n^2}.$$

*Proof.* To prove the first inequality, observe that

$$|Y - n| = |\max(Z, -n)| \leq |Z|.$$

Taking expectation yields the first equation. For the second term,

$$\frac{1}{Y} = \frac{1}{n} + \frac{n - Y}{Yn} = \frac{1}{n} + \frac{n - Y}{n^2} + \frac{(n - Y)^2}{n^2 Y} \leq \frac{1}{n} + \frac{n - Y}{n^2} + \frac{(n - Y)^2}{2n^2}. \tag{4}$$

Furthermore,

$$(n - Y)\mathbf{1}_{Y>0} = -Z\mathbf{1}_{Y>0} = -Z\mathbf{1}_{-Z<n} \leq -Z.$$

Combining the above two equations and using the fact that $|Y - n| \leq |Z|$ yields the second equation in the lemma. $\square$

# B Properties of the distance metric

*Proof of Lemma 1.* Recall that the distance between to histograms is given by

$$\ell_1(h_1, h_2) \stackrel{\text{def}}{=} \min_{D_1, D_2 : h(D_1) = h_1, h(D_2) = h_2} \ell_1(D_1, D_2)$$

$$= \min_{D_1, D_2 : h(D_1) = h_1, h(D_2) = h_2} \sum_{x \in \mathcal{X}} |n_x(D_1) - n_x(D_2)|.$$

Let $D_1^*$ and $D_2^*$ be the datasets that achieve the minimum above. Consider any two labels $x, y$ such that $n_x(D_1^*) \geq n_y(D_1^*)$. Let $D_2'$ be the dataset obtained as follows: $n_y(D_2') = n_x(D_2^*)$ and $n_x(D_2') = n_y(D_2^*)$ and for all other $z \notin \{x, y\}$, $n_z(D_2') = n_z(D_2^*)$. Since $D_2^*$ is the optimum,

$$\ell_1(D_1^*, D_2^*) \leq \ell_1(D_1^*, D_2').$$

Expanding both sides and canceling common terms, we get,

$$|n_x(D_1^*) - n_x(D_2^*)| + |n_y(D_1^*) - n_y(D_2^*)| \leq |n_x(D_1^*) - n_x(D_2')| + |n_y(D_1^*) - n_y(D_2')|$$
$$\leq |n_x(D_1^*) - n_y(D_2^*)| + |n_y(D_1^*) - n_x(D_2^*)|,$$

and thus if $n_x(D_1^*) \geq n_y(D_1^*)$, then $n_x(D_2^*) \geq n_y(D_2^*)$. Hence, the label of the $i^{th}$ highest count in both the datasets should be the same and

$$\ell_1(D_1^*, D_2^*) = \sum_{x \in \mathcal{X}} |n_x(D_1^*) - n_x(D_2^*)| = \sum_{i \geq 1} |n_{(i)}(h_1) - n_{(i)}(h_2)|.$$

$\square$

The distance measure satisfies triangle inequality, i.e., for any three histograms $h_1, h_2$, and $h_3$,

$$\ell_1(h_1, h_2) \leq \ell_1(h_1, h_3) + \ell_1(h_2, h_3).$$

The proof of the above equation is a simple consequence of Lemma 1 and is omitted. We now show that dividing histograms only increases the distance.

**Lemma 3.** *If $h = h_1 + h_2$ and $h' = h_1' + h_2'$, then*

$$\ell_1(h, h') \leq \ell_1(h_1, h_1') + \ell_1(h_2, h_2').$$

*Proof.* Since the elements in $h_1 + h_2$ are same as elements in $h$ and elements in $h_1' + h_2'$ are same as elements in $h_2$, there exists a permutation $\sigma$ such that

$$\ell_1(h_1, h_1') + \ell_1(h_2, h_2') = \sum_{i \geq 1} |n_{(i)}(h_1) - n_{(i)}(h_1')| + \sum_{i \geq 1} |n_{(i)}(h_2) - n_{(i)}(h_2')|$$

$$= \sum_{i \geq 1} |n_{(i)}(h) - n_{(\sigma_i)}(h')|.$$

Similar to proof of Lemma 1, it can be shown that the $\sigma$ that minimizes the above sum is the one that matches $i^{\text{th}}$ highest count in $h$ to $i^{\text{th}}$ highest count in $h'$ and hence

$$\ell_1(h, h') = \sum_{i \geq 1} |n_{(i)}(h) - n_{(i)}(h')| \leq \sum_{i \geq 1} |n_{(i)}(h) - n_{(\sigma_i)}|.$$

$\square$

It is useful to have few upper bounds on the $\ell_1$ distance over histograms.

**Lemma 4.** *For any two histograms $h_1, h_2$,*

$$\ell_1(h_1, h_2) \leq \sum_{r \geq 1}^{r_{\max}(h_1, h_2)} |\varphi_{r+}(h_1) - \varphi_{r+}(h_2)| \leq \sum_{r \geq 1} r|\varphi_r(h_1) - \varphi_r(h_2)|, \tag{5}$$

*where $r_{\max}(h_1, h_2)$ is the maximum $r$ such that $\varphi_r(h_1) + \varphi_r(h_2) > 0$.*

*Proof.* We prove the first inequality by induction on $r_{\max}(h_1, h_2)$. Suppose $r_{\max}(h_1, h_2) = 1$, then the inequality holds trivially as

$$\sum_i |n_{(i)}(h_1) - n_{(i)}(h_2)| = |\varphi_1(h_1) - \varphi_1(h_2)| = \sum_{r=1}^{r_{\max}(h_1,h_2)} |\varphi_{r+}(h_1) - \varphi_{r+}(h_2)|.$$

Now suppose it holds for all $r_{\max}(h_1, h_2) < r_0$. For $r_0 \stackrel{\text{def}}{=} r_{\max}(h_1, h_2)$. Let $h'_1$ and $h'_2$ be two datasets obtained as follows:

$$h'_i = \max(n_x, r_0 - 1) : n_x \in h_i\}.$$

This mapping preserves the ordering of $n_{(i)}$s up to ties and $r_{\max}(h'_1, h'_2) = r_0 - 1$. Thus,

$$\ell_1(h'_1, h'_2) \le \sum_{r=1}^{r_{\max}(h'_1,h'_2)} |\varphi_{r+}(h'_1) - \varphi_{r+}(h'_2)| = \sum_{r=1}^{r_{\max}(h_1,h_2)-1} |\varphi_{r+}(h_1) - \varphi_{r+}(h_2)|. \quad (6)$$

Hence,

$$\sum_i |n_{(i)}(h_1) - n_{(i)}(h_2)|$$

$$= \sum_{i:\max(n_{(i)}(h_1),n_{(i)}(h_2))<r_0} |n_{(i)}(h_1) - n_{(i)}(h_2)| + \sum_{i:\max(n_{(i)}(h_1),n_{(i)}(h_2))\ge r_0} |n_{(i)}(h_1) - n_{(i)}(h_2)|$$

$$= \sum_{i:\max(n_{(i)}(h_1),n_{(i)}(h_2))<r_0} |n_{(i)}(h'_1) - n_{(i)}(h'_2)|$$

$$+ \sum_{i:\max(n_{(i)}(h_1),n_{(i)}(h_2))\ge r_0} |n_{(i)}(h'_1) - n_{(i)}(h'_2) + \mathbf{1}_{n_{(i)}(h_1)=r_0} - \mathbf{1}_{n_{(i)}(h'_2)=r_0}|$$

$$\le \sum_{i:\max(n_{(i)}(h_1),n_{(i)}(h_2))<r_0} |n_{(i)}(h'_1) - n_{(i)}(h'_2)| + \sum_{i:\max(n_{(i)}(h_1),n_{(i)}(h_2))\ge r_0} |n_{(i)}(h'_1) - n_{(i)}(h'_2)|$$

$$+ \sum_{i:\max(n_{(i)}(h_1),n_{(i)}(h_2))\ge r_0} |\mathbf{1}_{n_{(i)}(h_1)=r_0} - \mathbf{1}_{n_{(i)}(h'_2)=r_0}|$$

$$= \ell_1(h'_1, h'_2) + |\varphi_{r_0}(h_1) - \varphi_{r_0}(h_2)|$$

Combining the above equation with Equation (6) yields the first inequality. For the second inequality, observe that

$$\sum_{r\ge 1}^{r_{\max}(h_1,h_2)} |\varphi_{r+}(h_1) - \varphi_{r+}(h_2)| = \sum_{r\ge 1}^{r_{\max}(h_1,h_2)} |\sum_{s\ge r} \varphi_s(h_1) - \varphi_s(h_2)|$$

$$\le \sum_{r\ge 1}^{r_{\max}(h_1,h_2)} \sum_{s\ge r} |\varphi_s(h_1) - \varphi_s(h_2)|$$

$$= \sum_{s\ge 1} s|\varphi_s(h_1) - \varphi_s(h_2)|,$$

where the inequality follows by triangle inequality and the last equality follows by observing that each term corresponding to index $s$ appears exactly $s$ times. $\square$

We now show a simple property of rounding off integers.

**Lemma 5.** *Let $x_1, x_2, \ldots, x_n$ be integers. Let $y_1, y_2, \ldots, y_n$ be real numbers. Let $\hat{y}_i$ be the nearest integer to $y_i$. Then,*

$$\sum_{i=1}^n |x_i - \hat{y}_i| \le 2 \sum_{i=1}^n |x_i - y_i|.$$

*Proof.* For any $i$,
$$|x_i - \hat{y}_i| \le |x_i - y_i| + |y_i - \hat{y}_i| \le 2|x_i - y_i|,$$
where the second inequality follows from the observation that $\hat{y}_i$ is the nearest integer to $y_i$. Summing over all indices $i$ yields the lemma. $\square$

We need the next auxilllary lemma, which we use in the proofs.

**Lemma 6.** *For a histogram $h_1$, let $h_1'$ be the histogram obtained by adding $k$ elements of value $t$ to $h$. Let $h_2'$ be another histogram and let $h_2$ is obtained by removing $k$ elements that are closest to $t$. Then*

$$\ell_1(h_1, h_2) \leq 2\ell_1(h_1', h_2').$$

*Proof.* Let $h_2''$ be the histogram obtained by adding $k$ elements of value $t$ to $h_2$. Since adding same number of elements to two datasets do not decrease the $\ell_1$ distance,

$$\ell_1(h_1, h_2) = \ell_1(h_1', h_2'') \leq \ell_1(h_1', h_2') + \ell_1(h_2', h_2''),$$

where the second inequality follows by triangle inequality. Consider the set of all histograms that have $\varphi_t \geq k$. Both $h_2''$ and $h_1'$ belong to this set. It can be shown that of all histograms in that set $h_2''$ is closest to $h_2'$ and hence

$$\ell_1(h_2', h_2'') \leq \ell_1(h_2', h_1'),$$

and hence the lemma. $\qquad\qquad\qquad\qquad\qquad\qquad\qquad\qquad\qquad\qquad\qquad\qquad\qquad\qquad\qquad\square$

## C   Privacy analysis of PRIVHIST

### C.1   Overview of privacy analysis

We break the analysis of PRIVHIST step by step. We will show that release of $N$ (step $(1)$) is $\epsilon_1$-DP. Then, we show that $H^{cs}$ and $H^{c\ell}$ are $\epsilon_2$-DP. Observe that PRIVHIST-LOWPRIVACY is just a post-processing step and by the post processing lemma does not need any differentialy privacy analysis. Finally we show that PRIVHIST-HIGHPRIVACY is $\epsilon_3$ differentially private. By the composition theorem [32], it follows that the total privacy cost is $\epsilon_1 + \epsilon_2 + \epsilon_3 = \epsilon$ and hence the privacy cost in Theorem 1. Of the above steps, proving $N$ is $\epsilon_1$-DP is straightforward and a sketch is in Lemma 7. Proving $H^{cs}$ and $H^{c\ell}$ is $\epsilon_2$-DP is more involved and is in Lemma 8. The main intuition behind Lemma 8 is a sensitivity preserving histogram split, which we describe below.

### C.2   Sensitivity preserving histogram split

Any two neighboring datasets $h_1$ and $h_2$ can fall into one of three categories:

1. They differ in $\varphi_T$ and $\varphi_{T+1}$.
2. They differ in $\varphi_r$ and $\varphi_{r+1}$ for some $0 \leq r < T - 1$.
3. They differ in $\varphi_r$ and $\varphi_{r+1}$ for some $r > T$.

For cases 2 and 3 above, it suffices to add noise to cumulative prevalences and counts as in **(3)** and **(4)**. However, if they differ in $\varphi_T$ and $\varphi_{T+1}$, the analysis is more involved. For example, consider the following simple example. $h_1 = \{\varphi_T = 1, \varphi_{n-T} = 1\}$ and $h_2 = \{\varphi_{T+1} = 1, \varphi_{n-T-1} = 1\}$. $h_1$ and $h_2$ have $\ell_1$ distance of one and are neighbors. If we divide $h_1$ in to two parts based on threshold $T$, say $h_1^s = \{\varphi_T = 1\}$ and $h_1^\ell = \{\varphi_{n-T} = 1\}$ and $h_2^s = \{\}$ and $h_2^\ell = \{\varphi_{T+1} = 1, \varphi_{n-T-1} = 1\}$, then $\ell_1(h_1^\ell, h_2^\ell) = T + 2$. Thus, if we naively add noise to cumulative prevalences for $r \leq T$ and to counts $r > T$, then we need to add noise $L(\mathcal{O}(T/\epsilon))$, which makes the utility of the algorithm much worse. To overcome this, we preprocess $h$ by moving $Z^b$ counts from $\varphi_T$ to $\varphi_{T+1}$, where $Z_b$ is a geometric random variable. This provides the required privacy without increasing the utility considerably. Finally, moving mass $Z^b$ can make the histogram to have negative prevalences. To overcome this, we add $M$ fake counts to $\varphi_T$ and $\varphi_{T+1}$.

### C.3   Technical details

We first prove a dataset depending composition theorem that helps us decompose differential privacy analysis depending on the dataset.

**Theorem 3** (Dataset dependent composition theorem)**.** *Let $Z_1, Z_2, \ldots, Z_n$ be a set of independent random variables. Let $X_1 = f_1(x, Z_1)$ be a deterministic function. Similarly let $X_i = f_i(X_{i-1}, Z_i)$ be deterministic functions for $2 \leq i \leq n$. If for any two neighboring data sets $x$ and $x'$,*

$$\min_{i \geq 1} \max_{z_1, z_2, \ldots, z_{i-1}, x_i} \frac{\Pr(X_i = x_i | x, z_1, z_2, \ldots, z_{i-1})}{\Pr(X_i = x_i | x', z_1, z_2, \ldots, z_{i-1})} \leq e^\epsilon.[3] \qquad (7)$$

*then $X_n$ is an $\epsilon$-DP output.*

*Proof.* For any two datasets, since $x \to X_1 \to X_n$ is a Markov chain,

$$\Pr(X_n = x_n | x) = \sum_{x_i} \Pr(X_n = x_n | X_i = x_i) \cdot \Pr(X_i = x_i | x).$$

Hence for any two datasets $x, x'$ any $x_n$, and for all $i$,

$$\frac{\Pr(X_n = x_n | x)}{\Pr(X_n = x_n | x')} = \frac{\sum_{x_1} \Pr(X_n = x_n | X_i = x_i) \cdot \Pr(X_i = x_i | x)}{\sum_{x_i} \Pr(X_n = x_n | X_i = x_i) \cdot \Pr(X_i = x_i | x')} \leq \max_{x_i} \frac{\Pr(X_i = x_i | x')}{\Pr(X_i = x_i | x)}.$$

Similarly for any $i$,

$$\Pr(X_i = x_i | x) = \sum_{z_1, z_2, \ldots, z_{i-1}} \Pr(X_i = x_i | x, z_1, z_2, \ldots, z_{i-1}) \cdot \Pr(z_1, z_2, \ldots, z_{i-1}).$$

Hence for any two datasets $x, x'$,

$$\frac{\Pr(X_i = x_i | x)}{\Pr(X_i = x_i | x')} = \frac{\sum_{z_1, z_2, \ldots, z_{i-1}} \Pr(X_i = x_i | x, z_1, z_2, \ldots, z_{i-1}) \cdot \Pr(z_1, z_2, \ldots, z_{i-1})}{\sum_{z_1, z_2, \ldots, z_{i-1}} \Pr(X_i = x_i | x', z_1, z_2, \ldots, z_{i-1}) \cdot \Pr(z_1, z_2, \ldots, z_{i-1})}$$

$$\leq \max_{z_1, z_2, \ldots, z_{i-1}} \frac{\Pr(X_i = x_i | x, z_1, z_2, \ldots, z_{i-1})}{\Pr(X_i = x_i | x', z_1, z_2, \ldots, z_{i-1})}.$$

Hence,

$$\max_{x_n} \frac{\Pr(X_n = x_n | x)}{\Pr(X_n = x_n | x')} \leq \min_{i \geq 1} \max_{x_i} \frac{\Pr(X_i = x_i | x')}{\Pr(X_i = x_i | x)}$$

$$\leq \min_{i \geq 1} \max_{x_i} \max_{z_1, z_2, \ldots, z_{i-1}} \frac{\Pr(X_i = x_i | x, z_1, z_2, \ldots, z_{i-1})}{\Pr(X_i = x_i | x', z_1, z_2, \ldots, z_{i-1})},$$

and hence for every pair of datasets if the right hand side is smaller than $e^\epsilon$, then $X_n$ is diffferentially private. □

## C.4 Privacy analysis

We start with proving $N$ is $\epsilon_1$-DP.

**Lemma 7.** *$N$ is $\epsilon_1$-DP.*

*Proof sketch.* The proof follows from Definition 3 and the fact that for any two neighboring datasets $h_1, h_2$, $n(h_1) - n(h_2) = |\sum_{n_x \in h_1} n_x - \sum_{n_x \in h_2} n_x| \leq \ell_1(h_1, h_2) = 1$, and hence the sensitivity of this query is 1. □

We now show that the release of $H^{cs}$ and $H^{c\ell}$ is DP.

**Lemma 8.** *Release of $H^{cs}$ and $H^{c\ell}$ is $\epsilon_2$-DP.*

*Proof.* $h \to H^b \to (H^{bs}, H^{b\ell}) \to (H^{cs}, H^{b\ell}) \to (H^{cs}, H^{c\ell})$ is a Markov chain. We use Theorem 3 to show that the output of this Markov chain is DP for all datasets. For any two neighboring datasets $h_1$ and $h_2$ can fall into one of three categories:

1. They differ in $\varphi_T$ and $\varphi_{T+1}$.

2. They differ in $\varphi_r$ and $\varphi_{r+1}$ for some $0 \leq r < T - 1$.

3. They differ in $\varphi_r$ and $\varphi_{r+1}$ for some $r > T$.

We prove that $(H^{cs}, H^{c\ell})$ release is $\epsilon_2$-DP for each of the above three cases.

**Case 1**: We show that the process $h \to H^b$ satisfies (7). Observe that

$$\max_{h^b} \frac{\Pr(H^b = h^b | h_2)}{\Pr(H^b = b^b | h_1)} \overset{(a)}{=} \frac{\Pr(\bar{\varphi}(H^b) = \bar{\varphi}^b | \bar{\varphi}(h_2))}{\Pr(\bar{\varphi}(H^b) = \bar{\varphi}^b | \bar{\varphi}(h_1))}$$

$$\overset{(b)}{=} \frac{\Pr(\varphi_T(H^b) = \varphi_T^b, \varphi_{T+1}(H^b) = \varphi_{T+1}^b | \varphi_T(h_2))}{\Pr(\varphi_T(H^b) = \varphi_T^b, \varphi_{T+1}(H^b) = \varphi_{T+1}^b | \bar{\varphi}_T(h_1))}$$

$$\overset{(c)}{=} \frac{\Pr(Z^b = \varphi_T^b - \varphi_T(h_2))}{\Pr(Z^b = \varphi_T^b - \varphi_T(h_1))}$$

$$= \frac{e^{-\epsilon_2 | \varphi_T^b - \varphi_T(h_2)|}}{e^{-\epsilon_2 | \varphi_T^b - \varphi_T(h_1)|}}$$

$$\leq e^{\epsilon_2 | \varphi_T(h_2) - \varphi_T(h_1)|}$$

$$\leq e^{\epsilon_2}.$$

$(a)$ follows by observing that there is a one to one correspondence between the histogram and the prevalences, $(b)$ follows from the fact that noise is added only to $\varphi_T$ and $\varphi_{T+1}$, and $(c)$ follows from the fact that the noise added to $\varphi_{T+1}$ is a deterministic function of noise added to $\varphi_T$, and $\varphi_T^b + \varphi_{T+1}^b = \varphi_T^a + \varphi_{T+1}^a$ always.

**Case 2**: We show that $(H^{bs}, H^{b\ell}) \to (H^{cs}, H^{b\ell})$ satisfies (7). Let $H_1^{bs}$ and $H_2^{bs}$ be the values of $H^{bs}$ for inputs $h_1$ and $h_2$ respectively. For any $Z_b = z_b$, since difference of maximums is at most maximum of differences,

$$\sum_r |\varphi_{r+}(H_2^{bs}) - \varphi_{r+}(H_1^{bs})| \leq \sum_r |\varphi_{r+}(h_2) - z_b - \varphi_{r+}(h_1) + z_b|$$

$$= \sum_r |\varphi_{r+}(h_2) - \varphi_{r+}(h_1)| \leq 1.$$

Hence,

$$\frac{\Pr((H^{cs}, H^{b\ell}) = (h^{cs}, h^{b\ell}) | H_2^{bs}, H^{b\ell})}{\Pr(H^{cs}, H^{b\ell}) = (h^{cs}, h^{b\ell}) | H_1^{bs}, H^{b\ell})} = \frac{\Pr(H^{cs} = h^{cs} | H_2^{bs})}{\Pr(H^{cs} = h^{cs} | H_1^{bs})}$$

$$= \frac{\Pr(\bar{\varphi}_{r+}(H^{cs}) = \bar{\varphi}_{r+}(h^{cs}) | \bar{\varphi}_{r+}(H_2^{bs}))}{\Pr(\bar{\varphi}_{r+}(H^{cs}) = \bar{\varphi}_{r+}(h^{cs}) | \bar{\varphi}_{r+}(H_1^{bs}))}$$

$$= \prod_{r \geq 1} \frac{\Pr(\varphi_{r+}(H^{cs}) = \varphi_{r+}(h^{cs}) | \bar{\varphi}_{r+}(H_2^{bs}))}{\Pr(\varphi_{r+}(H^{cs}) = \varphi_{r+}(h^{cs}) | \bar{\varphi}_{r+}(H_1^{bs}))}$$

$$= \prod_{r \geq 1} \frac{e^{-\epsilon_2 | \varphi_{r+}(h^{cs}) - \varphi_{r+}(H_2^{bs})|}}{e^{-\epsilon_2 | \varphi_{r+}(h^{cs}) - \varphi_{r+}(H_1^{bs})|}}$$

$$\leq \prod_{r \geq 1} e^{\epsilon_2 | \varphi_{r+}(H_2^{bs}) - \varphi_{r+}(H_1^{bs})|} \leq e^{\epsilon_2}.$$

The first equality follows from the observation that there is a one to one correspondence between $\varphi_r$s and $\varphi_{r+}$s. The second equation follows from the fact that noise added to various $\varphi_{r+}$s are independent of each other. The rest of the proof follows from the definition of the geometric mechanism and the fact that $h_1$ and $h_2$ are neighbors.

**Case 3**: We show that $(H^{cs}, H^{b\ell}) \to (H^{cs}, H^{c\ell})$ satisfies (7). Let $H_1^{b\ell}$ and $H_2^{b\ell}$ are the $H^{b\ell}$'s corresponding to $h_1$ and $h_2$ respectively. Conditioned on the value of $Z^b$, for any two neighboring histograms that differ in two consecutive $r$'s that are larger than $T$,

$$\sum_{r > T} \varphi_r(H_2^{b\ell}) - \sum_{r > T} \varphi_r(H_2^{b\ell}) = \sum_{r > T} \varphi_r(h_2) - \varphi_r(h_1) = 0.$$

Since their sums are equal, conditioned on the value of $Z^b$, it can be shown that $H_1^{b\ell}, H_2^{b\ell}$ are proper histograms and both contain $\max(0, M + Z^b + \sum_{r > T} \varphi_r(h_1))$ elements and they differ in at most two consecutive values of $r$. Thus $H_1^{b\ell}$ and $H_2^{b\ell}$ contain same number of counts and differ in at

most one count denoted by $i^*$. With these observations we now bound the ratio of probabilities for differential privacy.

Since there is a one to one correspondence between sorted counts and the histograms, we get

$$\frac{\Pr((H^{cs}, H^{c\ell}) = (h^{cs}, h^{c\ell})|H_2^{b\ell}, H^{cs})}{\Pr(H^{cs}, H^{c\ell}) = (h^{cs}, h^{c\ell})|H_1^{b\ell}, H^{cs})} = \frac{\Pr(H^{c\ell} = h^{c\ell}|H_2^{b\ell})}{\Pr(H^{c\ell} = h^{c\ell}|H_1^{b\ell})}$$

$$= \frac{\prod_i \Pr(N_i = n_i|n_{(i)}(H_2^{b\ell}))}{\prod_i \Pr(N_i = n_i|n_{(i)}(H_2^{b\ell}))}$$

$$= \frac{\Pr(N_{i^*} = n_{i^*}|n_{(i^*)}(H_2^{b\ell}))}{\Pr(N_{i^*} = n_{i^*}|n_{(i^*)}(H_1^{b\ell}))}$$

$$= \frac{e^{-\epsilon_2|n_{i^*} - n_{(i^*)}(H_2^{b\ell})|}}{e^{-\epsilon_2|n_{i^*} - n_{(i^*)}(H_1^{b\ell})|}} \le e^{\epsilon_2|n_{(i^*)}(H_2^{b\ell}) - n_{(i^*)}(H_1^{b\ell})|} \le e^{\epsilon_2},$$

where the last set of inequalities follow from the definition of geometric mechanism and the fact that the noise added to $N_{(i)}$s are independent of each other, and $n_{(i^*)}$ is the only count in which the two histograms differ. $\qquad \square$

We now show that PRIVHIST-HIGPRIVACY is $\epsilon_3$-DP.

**Lemma 9.** *The output of* PRIVHIST-HIGPRIVACY *is $\epsilon_3$-DP.*

*Proof.* Observe that $h \to H^v \to H^w \to H^x \to H^y$ is a Markov chain, hence it suffices to prove $H^v \to H^w$ is $\epsilon_3$-DP.

Let $h_1$ and $h_2$ are two neighboring datasets. Without loss of generality, let they differ in $\varphi_k$ and $\varphi_{k+1}$. Let $h_1^v$ and $h_2^v$ be the two histograms obtained after quanitzation step (3). Since $h_1$ and $h_2$ differ only at $k$ and $k+1$, $h_1^v$ and $h_2^v$ differ only in $\varphi_{s_{i-1}}^v$ and $\varphi_{s_i}^v$ for some $i$. Furthermore,

$$|\varphi_{s_{i-1}}^v(h_1^v) - \varphi_{s_{i-1}}^v(h_2^v)| = |\varphi_{s_i}^v(h_1^v) - \varphi_{s_i}^v(h_2^v)| \le \frac{1}{s_i - s_{i-1}}.$$

For all $j \notin \{s_{i-1}, s_i\}$, $\varphi_j^v(h_1^v) = \varphi_j^v(h_2^v)$. Hence $\varphi_{s_i+}^v(h_1^v) \ne \varphi_{s_i+}^v(h_2^v)$ for only one value of $i$, let $i^*$ be this value.

$$\frac{\Pr(H^w = h_w|h_1^v)}{\Pr(H^w = h_w|h_2^v)} = \prod_i \frac{\Pr(\varphi_{s_i+}(H^w) = \varphi_{s_i+}(h^w)|\varphi_{s_i+}(h_1^v))}{\Pr(\varphi_{s_i+}(H^w) = \varphi_{s_i+}(h^w)|\varphi_{s_i+}(h_2^v))}$$

$$= \frac{\Pr(\varphi_{s_i^*+}(H^w) = \varphi_{s_i^*+}(h^w)|\varphi_{s_i^*+}(h_1^v))}{\Pr(\varphi_{s_i^*+}(H^w) = \varphi_{s_i^*+}(h^w)|\varphi_{s_i^*+}(h_2^v))}$$

$$= \frac{\Pr(Z^{s_{i^*}} = \varphi_{s_i^*+}(h^w) - \varphi_{s_{i^*+}}(h_1^v))}{\Pr(Z^{s_{i^*}} = \varphi_{s_i^*+}(h^w) - \varphi_{s_{i^*+}}(h_2^v))}$$

$$\le \exp\left(\frac{\epsilon_3(s_{i^*} - s_{i^*-1})}{s_{i^*} - s_{i^*-1}}\right) \le e^{\epsilon_3}.$$

$\qquad \square$

# D  Utility analysis of PRIVHIST

Let $H^o$ be the output of either PRIVHIST-LOWPRIVACY or PRIVHIST-HIGHPRIVACY. In both the low and high privacy regimes, the output error can be bounded as

$$\ell_1(h, H^o)\mathbf{1}_{N>0} + n\mathbf{1}_{N \le 0}.$$

Furthermore,

$$\mathbb{E}[n\mathbf{1}_{N \le 0}] \le ne^{-n\epsilon_1} \lesssim e^{-\epsilon_1/2}, \text{ [4]}$$

where the last inequality, follows by breaking it in to cases $n = 0$ and $n > 0$. The bound on $\mathbb{E}[|N - n|]$ follows from the fact that $N = n + G(e^{-\epsilon_1})$, Lemma 2, and the moments of the geometric distribution. In the next two sections, we bound $\ell_1(h, H^o)\mathbf{1}_{N>0}$ for both low privacy and high privacy regimes.

## D.1 Low privacy regime

In the following analysis, let $Z$ be a geometric random variable distributed as $G(e^{-\epsilon_2})$. Let $H^{b'} = H^{bs} + H^{b\ell}$. For the bounds on the histogram, by Lemma 6 and triangle inequality,

$$\ell_1(h, H^e) \leq 4\ell_1(H^a, H^d) \leq 4\ell_1(H^{b'}, H^d) + 4\ell_1(H^{b'}, H^a). \tag{8}$$

For the second term, observe that since $H^{b'}$ is obtained by moving $Z^b$ terms between $T$ and $T+1$ and then majorizing it. If $|Z^b| \leq M$, then majorization does not change the histogram. Hence,

$$\ell_1(H^{b'}, H^a) = \ell_1(H^{b'}, H^a)\mathbf{1}_{|Z^b| \leq M} + \ell_1(H^{b'}, H^a)\mathbf{1}_{|Z^b| > M} \leq |Z^b| + (n + 2M(T+1))\mathbf{1}_{|Z^b| > M}.$$

Taking expectation on both sides

$$\mathbb{E}_N[\ell_1(H^{b'}, H^a)] \leq \mathbb{E}[|Z^b| + (n + 2M(T+1))\mathbf{1}_{|Z^b| > M}] \lesssim \mathbb{E}[Z] + \frac{n+N}{N^2}e^{-2\epsilon_2} \tag{9}$$

For the first term, by Lemma 3,

$$\ell_1(H^{b'}, H^d) \leq \ell_1(H^{bs}, H^{ds}) + \ell_1(H^{b\ell}, H^{d\ell}). \tag{10}$$

We now bound both the terms above. For the large counts, let $i_j$ be the index of the noisy count $N_{(i)}(H^{b\ell})$.

$$\ell_1(H^{b\ell}, H^{d\ell}) \leq \sum_i |N_{(i)}(H^{b\ell}) - N_{(i)}(H^{d\ell})|$$

$$\leq \sum_i |N_{(i)}(H^{b\ell}) - N_{i_j}(H^{d\ell})|$$

$$\leq \sum_i |N_{(i)}(H^{b\ell}) - N_{i_j}(H^{c\ell})|.$$

Since the number of terms above $T$ is at most $n/T + M + Z^b$, in expectation,

$$\mathbb{E}_N[\ell_1(H^{b\ell}, H^{d\ell})] \leq \mathbb{E}\left[\sum_i |Z_i^{c\ell}|\right] \lesssim \left(\frac{n}{T} + M + \mathbb{E}|Z|\right) \cdot \mathbb{E}[|Z|]. \tag{11}$$

For the smaller counts observe that

$$\ell_1(H^{bs}, H^{ds}) \leq \sum_{r \geq 0} |\varphi_{r+}^{bs} - \varphi_{r+}^{ds}|$$

$$\overset{(a)}{\leq} 2\sum_{r \geq 0} |\varphi_{r+}^{bs} - \varphi_{r+}^{\text{mon}}|$$

$$\overset{(b)}{\leq} 2\sqrt{T} \cdot \sqrt{\sum_{r \geq 0}(\varphi_{r+}^{bs} - \varphi_{r+}^{\text{mon}})^2}$$

$$\overset{(c)}{\leq} 2\sqrt{T} \cdot \sqrt{\sum_{r \geq 0}(\varphi_{r+}^{bs} - \varphi_{r+}^{\text{cs}})^2}$$

$$\leq 2\sqrt{T} \cdot \sqrt{\sum_{r \geq 0}(Z_r^{cs} - Z^b)^2}, \tag{12}$$

where $(a)$ follows from the fact that rounding off increases the error at most by 2 (Lemma 5), $(b)$ follows by the Cauchy-Schwarz inequality, and $(c)$ follows from the fact that $\varphi_{r+}^{bs}$ are monotonic and hence monotonic projection only decreases the error. Hence in expectation,

$$\mathbb{E}_N[\ell_1(H^{bs}, H^{ds})] \lesssim T\sqrt{\mathbb{E}[Z^2]}. \tag{13}$$

Combining (8), (9), (10), (11), (12), and (13),

$$\mathbb{E}_N[\ell_1(h, H^e)] \lesssim |Z| + 4T\sqrt{\mathbb{E}[Z^2]} + 2\left(\frac{n}{T} + M + \mathbb{E}|Z|\right) \cdot \mathbb{E}[|Z|] + \frac{n+N}{N^2}e^{-2\epsilon_2}.$$

Substituting $T = \lceil\sqrt{N}\rceil$ and $M = \lceil\frac{2\log N\epsilon_2}{\epsilon_2}\rceil$, yields

$$\mathbb{E}_N[\ell_1(h, H^e)] \lesssim |Z| + \sqrt{N}\sqrt{\mathbb{E}[Z^2]} + 2\left(1 + \frac{n}{\sqrt{N}} + \frac{\log N\epsilon_2}{\epsilon_2} + \mathbb{E}|Z|\right) \cdot \mathbb{E}[|Z|] + \frac{n+N}{N}e^{-2\epsilon_2}.$$

Taking expectation w.r.t. $N$ and using Lemma 2 yields

$$\mathbb{E}[\ell_1(h, H^e)\mathbf{1}_{N>0}] \lesssim \sqrt{n\mathbb{E}[Z^2]} + e^{-2\epsilon_2} \lesssim \sqrt{n}e^{-\epsilon/6},$$

where the last inequality follows from moments of geometric distribution.

## D.2 High privacy utility

For the high privacy regime, by the triangle inequality,

$$\ell_1(h, H^y) \le \ell_1(h, H^u) + \ell_1(H^u, H^y). \tag{14}$$

For the first term, since we are only reducing counts of certain elements,

$$\mathbb{E}[\ell_1(h, H^u)] \le \mathbb{E}[\mathbf{1}_{N<n/2}n] \le e^{-n\epsilon_1/2}n \lesssim \frac{1}{\epsilon_1}. \tag{15}$$

The second term can be bounded as

$$\mathbb{E}[\ell_1(H^u, H^y)] \le \sum_{r>0}|\varphi_{r+}(H^u) - \varphi_{r+}(H^y)|$$

$$\overset{(a)}{\le} 2\sum_{r>0}|\varphi_{r+}(H^u) - \varphi_{r+}(H^x)|$$

$$\le 2\sum_{r>0}|\varphi_{r+}(H^u) - \varphi_{r+}(H^v)| + |\varphi_{r+}(H^x) - \varphi_{r+}(H^v)|, \tag{16}$$

where $(a)$ follows by Lemma 5 and the last inequality follows by the triangle inequality. The first term in the last equation corresponds to the smoothing error and we analyze it now.

$$\varphi^v_{s_{i+1}+} = \sum_{r=s_i}^{s_{i+1}}\varphi^u_r\frac{(r-s_i)}{s_{i+1}-s_i} + \sum_{r>s_{i+1}}\varphi^u_r.$$

Since $\varphi^v_j = 0$ for $j \notin S$,

$$\sum_{j=s_i}^{s_{i+1}-1}|\varphi^v_{j+} - \varphi^u_{j+}| = \sum_{j=s_i}^{s_{i+1}-1}\left|\sum_{r=s_i}^{s_{i+1}}\varphi^u_r\frac{(r-s_i)}{s_{i+1}-s_i} - \sum_{r=j}^{s_{i+1}}\varphi^u_r\right|$$

$$\le \sum_{j=s_i}^{s_{i+1}}\sum_{r=s_i}^{s_{i+1}}\varphi^u_r\left|\frac{(r-s_i)}{s_{i+1}-s_i} - \mathbf{1}_{r\ge j}\right|$$

$$= \sum_{r=s_i}^{s_{i+1}}2\varphi^u_r\frac{(r-s_i)(s_{i+1}-r)}{s_{i+1}-s_i}$$

$$\le 2\sum_{r=s_i}^{s_{i+1}}\varphi^u_r\min(s_{i+1}-r, r-s_i).$$

For $r \le T'$, that lies between $s_i$ and $s_{i+1}$ in $S$,

$$|r - s_i| \le \min(\lfloor T(1+q)^{i+1}\rfloor - r, r - \lfloor T(1+q)^i\rfloor) \lesssim rq.$$

If $r \geq T'$, the analysis depends on the value of $Z_b$. If $Z_b \geq -M$, $r \geq T'$, and $\varphi_r^u > 0$, then there exists a $s_i$ that is at most $Z_i^{cl}$ away for from $r$. If not, then the error is at most $2n$. Hence, the smoothing error in expectation can be bounded by

$$\lesssim \sum_{r \geq T}^{T'} \varphi_r^u qr + \sum_{r \geq T'} \varphi_r^u \mathbb{E}_N[|Z| \mathbf{1}_{Z_b \geq -M}] + \mathbb{E}_N[2n\mathbf{1}_{Z_b < -M}] \lesssim nq + \frac{n\mathbb{E}[|Z|]}{T'} + \frac{n}{N}. \quad (17)$$

For the second part,

$$\sum_{r \geq 1} |\varphi_{r+}^v - \varphi_{r+}^x| \overset{(a)}{\leq} \sum_{s_i \in S} |\varphi_{s_i+}^v - \varphi_{s_i+}^x|(s_i - s_{i-1})$$

$$\overset{(b)}{\leq} \sqrt{|S|} \cdot \sqrt{\sum_{s_i \in S} (\varphi_{s_i+}^v - \varphi_{s_i+}^x)^2 (s_i - s_{i-1})^2}$$

$$\overset{(c)}{\leq} \sqrt{|S|} \cdot \sqrt{\sum_{s_i \in S} (\varphi_{s_i+}^w - \varphi_{s_i+}^x)^2 (s_i - s_{i-1})^2},$$

where $(a)$ follows by observing that $\varphi_r^x = 0$ for $r \notin S$, $(b)$ follows from the Cauchy-Schwarz inequality. $(c)$ follows from the fact that projecting on to the simplex only increases the error. By the second moments of $Z_i^s$, taking expectation on both sides yields

$$\sum_{r \geq 1} \mathbb{E}_N|\varphi_{r+}^v - \varphi_{r+}^x| \lesssim |S| \cdot \frac{1}{\epsilon}.$$

We now bound the size of the set $S$. By step $(2)$ of the algorithm, number of elements in $S$ can be bounded by

$$\lesssim T + 1 + \log_{1+q} \frac{T'}{T} + \frac{10n}{T'} + U,$$

where $U$ is the number of elements less than $T'/10$ in $H^{b\ell}$, that upon adding noise increases to $T'$. Expectation of $U$ conditioned on $N$ is

$$\mathbb{E}_N[U] \lesssim \left(\frac{n}{T} + M + \mathbb{E}[|Z|]\right) e^{-9T'\epsilon_2/10} \lesssim \left(\frac{n}{\sqrt{N\epsilon}} + 1 + \frac{\log N}{\epsilon}\right) e^{-3\sqrt{N/\epsilon}} \lesssim \frac{n\epsilon}{N} + 1, \quad (18)$$

where the second inequality follows by substituting $T' = 10\sqrt{N/\epsilon_3^3}$ and third inequality follows by algebraic manipulation. Combining (14), (15), (16), (17), and (18), and using the fact that quantization error is at most $\mathcal{O}(n)$ yields

$$\lesssim \min(nq, n) + \frac{n\mathbb{E}[|Z|]}{T'} + \left(T + 1 + \log_{1+q} \frac{T'}{T} + \frac{n}{T'} + \frac{n\epsilon}{N}\right) \frac{1}{\epsilon} + \frac{n}{N}$$

$$\lesssim \min(nq, n) + \left(\sqrt{N\epsilon} + \frac{1}{q} \log \frac{2}{\epsilon} + \frac{n\sqrt{\epsilon^3}}{\sqrt{N}} + \frac{n\epsilon}{N}\right) \frac{1}{\epsilon} + \frac{n}{N}$$

$$\lesssim n \min\left(\frac{1}{\sqrt{N}} \sqrt{\frac{1}{\epsilon} \log \frac{2}{\epsilon}}, 1\right) + \left(\sqrt{N} + 1\right) \sqrt{\frac{1}{\epsilon} \log \frac{2}{\epsilon}} + \frac{n\sqrt{\epsilon}}{\sqrt{N}} + \frac{n}{N},$$

where the last equation follows by substituting the value of $q = \sqrt{\frac{1}{N\epsilon_3} \log \frac{1}{\epsilon_3}}$. Taking expectation with respect to $N$ and using Lemma 2, (4) and the fact that $n\epsilon \gtrsim 1$ yields,

$$\mathbb{E}[\ell_1(h, H^z)\mathbf{1}_{N>0}] \lesssim \sqrt{\frac{n}{\epsilon} \cdot \log \frac{2}{\epsilon}} + \frac{1}{\epsilon}.$$

### D.3 Time complexity

In this section, we provide a proof sketch of the time complexity. Suppose $\epsilon > 1$. The number of prevalences in $H^{bs}$ is $T \lesssim \sqrt{N}$. Similarly, the number of counts in $H^{b\ell}$ is $n/T \approx n/\sqrt{N}$. Further,

the number of additional fake counts added is $M \lesssim \log N$ and the isotonic regression using PAVA takes linear in the size of the input. Hence, the expected run time, conditioned on $N$ is at most $\sqrt{N} + \frac{n}{\sqrt{N}} + M$, which upon taking expectation w.r.t. $N$ yields an expected run time of $\sqrt{n}$.

If $\epsilon \leq 1$, steps (1)-(4) in PRIVHIST takes time $\lesssim \sqrt{N/\epsilon} + \log N/\epsilon$. The time complexity to find boundaries, smooth prevalences, add noise, and perform isotonic regression is $|S| + \sqrt{n}$. By the utility analysis, $|S| \lesssim \sqrt{N \log \frac{1}{\epsilon}} + \frac{n\sqrt{\epsilon}}{\sqrt{N}}$. Summing all the time complexities and taking expectation w.r.t. $N$ similar to the utility analysis, yields a total time complexity of $\tilde{\mathcal{O}}\left(\sqrt{\frac{n}{\epsilon}} + \frac{\log n}{\epsilon}\right)$.

# E   Lower bound in the low privacy regime

**Lemma 10.** *Let $x \in \{0,1\}^k$. Suppose two vectors $x, y$ are neighbors if $||x - y||_1 \leq 1$. If $\hat{X} = M(x)$ be an $\epsilon$-DP estimate of $x$, then*

$$\max_{x \in \{0,1\}^k} \mathbb{E}[||x - \hat{X}||_1] \gtrsim ke^{-\epsilon}.$$

*Proof.* Let $p(x)$ be the uniform distribution over $\{0,1\}^k$. For a vector $v \in \{0,1\}^{k-1}$, let $\mathcal{X}_v^i$ denote the two vectors such that $x_1^{i-1} = v_1^{i-1}$ and $x_{i+1}^k = v_i^{k-1}$. Then

$$\max_{x \in \{0,1\}^k} \mathbb{E}[||x - \hat{X}||_1] \geq \frac{1}{2^k} \sum_{x \in \{0,1\}^k} \mathbb{E}[||x - \hat{X}||_1]$$

$$= \frac{1}{2^k} \sum_{x \in \{0,1\}^k} \sum_{i=1}^k \mathbb{E}|x_i - \hat{X}_i|$$

$$= \sum_{i=1}^k \frac{1}{2^k} \sum_{x \in \{0,1\}^k} \mathbb{E}|x_i - \hat{X}_i|$$

$$= \sum_{i=1}^k \frac{1}{2^{k-1}} \sum_{v \in \{0,1\}^{k-1}} \frac{1}{2} \sum_{x \in \mathcal{X}_v^i} \mathbb{E}|x_i - \hat{X}_i|. \tag{19}$$

For any $v$ and $i$, vectors in $\mathcal{X}_v^i$ are neighbors and hence by the definition of DP,

$$\sum_{x \in \mathcal{X}_v^i} \mathbb{E}|x_i - \hat{X}_i| = \sum_{\hat{x}_i} \Pr(\hat{X}_i = \hat{x}_i | x_i = 1)|1 - \hat{x}_i| + \Pr(\hat{X}_i = \hat{x}_i | x_i = 0)|0 - \hat{x}_i|$$

$$\geq \sum_{\hat{x}_i} \Pr(\hat{X}_i = \hat{x}_i | x_i = 1)|1 - \hat{x}_i| + e^{-\epsilon} \Pr(\hat{X}_i = \hat{x}_i | x_i = 1)|0 - \hat{x}_i|$$

$$\geq e^{-\epsilon} \sum_{\hat{x}_i} \Pr(\hat{X}_i = \hat{x}_i | x_i = 1)$$

$$\geq e^{-\epsilon}.$$

Substituting the above lower bound in (19) yields the lemma. $\qquad \square$

We now use the above bound to prove Theorem 2. We first define a of histograms to show the lower bound. Let $k = \sqrt{n}/10$. For a given vector $x \in \{0,1\}^k$, let $\varphi_{4i} = \varphi_{4i+3} = x_{i-1}$ and $\varphi_{4i+1} = \varphi_{4i+2} = 1 - x_{i-1}$, $\varphi_r = 1$ for $r = n - \sum_{i=1}^{4k} \varphi_r r$, and $\varphi_r = 0$ otherwise. Observe that $\sum_{r \geq 1} \varphi_r r = n$ and are valid histograms. If two vectors $x$ and $y$ have hamming distance $d$, then the corresponding distance between anonymized histograms is $2d$.

Consider a slightly different definition of neighboring datasets over histograms, where two datasets are neighboring if the neighboring datasets are distance 2 apart. If a mechanism is $\epsilon$-DP in the previous definition of neighboring datasets, then the mechanism is $2\epsilon$-DP in the new notion of neighbors.

One mechanism for releasing the vectors $x \in \{0,1\}^k$ with $2\epsilon$-DP is to encode it as the histograms as mentioned above and release them. Since such a mechanism has hamming distance $\gtrsim ke^{-2\epsilon}$, the

anonymized histograms cannot be estimated with accuracy $\gtrsim ke^{-2\epsilon}$ with $2\epsilon$-DP under new definition of neighbors and hence it cannot be estimated with accuracy $\gtrsim ke^{-2\epsilon}$ with $\epsilon$-DP under the old definition of neighbors.

# F  Proof of Corollary 1

Recall that $N$ is the DP estimate of $n$ and $H$ is the DP estimate of $h$. In the following Let $X^n$ be the initial set of $n$ samples. Let $X^N$ be $N$ samples obtained from $p$, $X^n$ as follows. If $N < n$, obtain $X^N$ by removing $n - N$ samples uniformly from $X^n$. If $N \geq n$, add $N - n$ samples from $p$ to $X^n$. Note that $X^N$ are $N$ i.i.d. samples from $p$. We bound the error of the estimator as follows.

$$f(p) - \hat{f}^p = (f(p) - \hat{f}^p)\mathbf{1}_{N>0} + (f(p) - \hat{f}^p)\mathbf{1}_{N=0}.$$

Taking expectation on the second term,

$$\mathbb{E}[(f(p) - \hat{f}^p)\mathbf{1}_{N=0}] = \mathbb{E}[f(p)\mathbf{1}_{N=0}] \leq f(p)e^{-nc\epsilon},$$

for some constant $c$. For the case $N > 0$, we bound as follows.

$$
\begin{aligned}
&f(p) - \hat{f}^p \\
&= f(p) - \sum_{r \geq 1} f(r, N)\varphi_r(H) \\
&= f(p) - \sum_{r \geq 1} f(r, N)\varphi_r(h(X^N)) + \sum_{r \geq 1} f(r, N)\varphi_r(h(X^N)) - \sum_{r \geq 1} f(r, N)\varphi_r(h(X^n)) \\
&\quad + \sum_{r \geq 1} f(r, N)\varphi_r(h(X^n)) - \sum_{r \geq 1} f(r, N)\varphi_r(H).
\end{aligned}
$$

We bound each of the three (difference) terms above. First observe that by the assumptions in the theorem:

$$\mathbb{E}_N\left[\left| f(p) - \sum_{r \geq 1} f(r, N)\varphi_r(h(X^N)) \right|\right] \leq \mathcal{E}(\hat{f}, N) \leq \mathcal{E}(\hat{f}, n) + |N - n|\frac{N^\beta}{N}.$$

Since $X^N$ and $X^n$ differ in at most $N - n$ terms, by the properties of sorted $\ell_1$ distance,

$$\sum_{r \geq 1} f(r, N)\varphi_r(h(X^N)) - \sum_{r \geq 1} f(r, N)\varphi_r(h(X^n)) \leq |N - n| \max_r |f(r, N)| \leq |N - n| \cdot \frac{N^\beta}{N}.$$

Further by the properties of sorted $\ell_1$ distance,

$$\mathbb{E}_N[|\sum_{r \geq 1} f(r, N)\varphi_r(h(X^n)) - \sum_{r \geq 1} f(r, N)\varphi_r(H)|] \leq \max_r |f(r, N)| \mathbb{E}_N[\ell_1(h, H)]$$

$$\lesssim \frac{N^\beta}{N} \cdot \mathbb{E}_N[\ell_1(h, H)].$$

Summing all the three terms, we get that the error is at most

$$\lesssim \mathcal{E}(\hat{f}, n) + |N - n| \cdot \frac{N^\beta}{N} + \frac{N^\beta}{N} \cdot \mathbb{E}_N[\ell_1(h, H)]. \tag{20}$$

For $\epsilon > 1$, difference between the expected error and $\mathcal{E}(\hat{f}, n)$ is

$$\lesssim \mathbb{E}\left[\frac{N^\beta}{N}|N - n|\mathbf{1}_{N>0} + \frac{N^\beta}{N}\left(\sqrt{N} + \frac{n}{\sqrt{N}}\right)\mathbf{1}_{N>0}\right]e^{-c\epsilon}$$

$$\lesssim \mathbb{E}\left[\frac{N^\beta}{N}|N - n|\mathbf{1}_{N>n/2} + \frac{N^\beta}{N}\left(\sqrt{N} + \frac{n}{\sqrt{N}}\right)\mathbf{1}_{N>n/2}\right]e^{-c\epsilon} + \mathbb{E}\left[n\mathbf{1}_{n/2\geq N>0}\right]e^{-c\epsilon}$$

$$\lesssim \mathbb{E}\left[\frac{n^\beta}{n}|N - n| + \frac{n^\beta}{\sqrt{n}}\right]e^{-c\epsilon} + \mathbb{E}\left[n\mathbf{1}_{n/2\geq N>0}\right]e^{-c\epsilon}$$

$$\lesssim n^{\beta-1/2}e^{-c\epsilon} + ne^{-nc'\epsilon}$$

$$\lesssim n^{\beta-1/2}e^{-c\epsilon},$$

where the last inequality uses the fact that $\epsilon > 1$. Combining the result with the case $N = 0$, we get that

$$\mathbb{E}[|f(p) - \hat{f}^p|] \lesssim \mathcal{E}(\hat{f}, n) + n^{\beta - 1/2} e^{-c\epsilon} + f(p) e^{-n\epsilon}.$$

The result for $\epsilon > 1$ follows if

$$n \geq \max\left( n(\hat{f}, \alpha), \mathcal{O}\left( \left( \frac{1}{\alpha e^{c\epsilon}} \right)^{\frac{2}{1-2\beta}} + \frac{1}{\epsilon} \log \frac{f_{\max}}{\alpha} \right) \right).$$

For $\Omega(1/n) \leq \epsilon \leq 1$, by (20), the difference between the expected error and $\mathcal{E}(\hat{f}, n)$ is

$$\lesssim \mathbb{E}\left[ \frac{N^\beta}{N} \left( |N - n| + n \min\left( \frac{1}{\sqrt{N}} \sqrt{\frac{1}{\epsilon} \log \frac{2}{\epsilon}}, 1 \right) + \sqrt{N} \sqrt{\frac{1}{\epsilon} \log \frac{2}{\epsilon}} + \frac{n\sqrt{\epsilon}}{\sqrt{N}} + \frac{n}{N} \right) \mathbf{1}_{N>0} \right]$$

$$\lesssim \mathbb{E}\left[ \frac{N^\beta}{N} \left( |N - n| + \sqrt{N} \sqrt{\frac{1}{\epsilon} \log \frac{2}{\epsilon}} + \frac{n\sqrt{\epsilon}}{\sqrt{N}} + \frac{n}{N} \right) \mathbf{1}_{N>n/2} \right] + \mathbb{E}\left[ \left( \sqrt{\frac{1}{\epsilon} \log \frac{2}{\epsilon}} + n \right) \mathbf{1}_{n/2 \geq N>0} \right]$$

$$\lesssim \mathbb{E}\left[ \frac{n^\beta}{n} |N - n| + \frac{n^\beta}{n} \left( \sqrt{n} \sqrt{\frac{1}{\epsilon} \log \frac{2}{\epsilon}} \right) \right] + \mathbb{E}\left[ \left( \sqrt{\frac{1}{\epsilon} \log \frac{2}{\epsilon}} + n \right) \mathbf{1}_{n/2 \geq N>0} \right]$$

$$\lesssim \frac{n^\beta}{n\epsilon} + \frac{n^\beta}{\sqrt{n}} \sqrt{\frac{1}{\epsilon} \log \frac{2}{\epsilon}} + \left( \sqrt{\frac{1}{\epsilon} \log \frac{2}{\epsilon}} + n \right) e^{-nc\epsilon}$$

$$\lesssim \frac{n^\beta}{\sqrt{n}} \sqrt{\frac{1}{\epsilon} \log \frac{2}{\epsilon}} + n e^{-nc\epsilon}.$$

Combining with the result for the case $N = 0$, we get

$$\mathbb{E}[|f(p) - \hat{f}^p|] \lesssim \mathcal{E}(\hat{f}, n) + n^{\beta - 1/2} \sqrt{\frac{1}{\epsilon} \log \frac{2}{\epsilon}} + (n + f(p)) e^{-nc\epsilon}.$$

The result for $\Omega(1/n) \leq \epsilon < 1$ follows if

$$n \geq \max\left( n(\hat{f}, \alpha), \mathcal{O}\left( \left( \frac{\sqrt{\log(2/\epsilon)}}{\alpha\sqrt{\epsilon}} \right)^{\frac{2}{1-2\beta}} + \frac{1}{\epsilon} \log \frac{f_{\max}}{\alpha\epsilon} \right) \right).$$

## Footnotes

[3] For notational simplicity, let $z_1^0 = \emptyset$.

[4] We use $\lesssim$ instead of $\mathcal{O}$ notation and $\gtrsim$ instead of $\Omega$ notation for compactness.

[5]$\mathbb{E}_N$ denotes conditional expectation w.r.t. $N$