[Reviews · NeurIPS 2019]

Reviewer 1



This paper considers the problem of privately estimating the "fingerprint" of a dataset. A dataset over some universe consists of (x,n_x) pairs, where x is a label, and n_x is the number of times that label x occurs. The fingerprint of this dataset is the histogram over the n_x's: it discards the labels, and, for each r >= 0, it counts the number of labels which were observed exactly r times. This is equivalent to approximating the histogram of the distribution up to a permutation. Under some appropriate distance measure (the minimum L1 distance over datasets which may correspond to the given fingerprints), this paper gives an algorithm which estimates the distribution up to accuracy ~O(sqrt(n/epsilon)) (in the high-privacy regime -- in the low-privacy regime, there is also a O_epsilon(sqrt(n)) upper bound). There is a known lower bound of Omega(sqrt(n/epsilon)) in the high-privacy regime, showing that the upper bound here is optimal up to a log(1/epsilon) factor. There is a new lower bound in this paper for the low-privacy regime, which is close to tight (up to a constant in the exponent multiplying epsilon -- as such, the use of "near-optimal" is slightly imprecise in line 119-120, as I usually read this as optimal up to polylogarithmic factors). The authors also provide a corollary of their main upper bound, to estimating symmetric properties of distributions. The results seem quite strong. It is impressive that this paper manages to essentially settle the problem in both the high and low privacy regimes. (Very) naive methods would give error of O(n/eps) (the authors should provide a comparison with the sample complexity for the problem of estimating the histogram itself, which I believe is this same value of n/eps), this paper improves the accuracy quadratically in both n and eps. The algorithm itself is fairly "elementary" -- there are a lot of steps, but pretty much all of them are simple operations, which may result in some adaptation of this method being practically realizable. There is a rather careful "partitioning" method into high and low count items in order to achieve the sqrt(n) error bound, rather than the naive n. My biggest criticism with this work is perhaps the clarity of the actual method. While a prefix of the paper is rather well-written and provides a nice exposition, unfortunately, I found the technical core of the actual result difficult to follow. The pseudocode itself (page 8) is rather inscrutable by itself, so one must consult the accompanying prose description (Section 4). However, the critical portions of this (Sections 4.2 and 4.3) are a bit terse, and marred by typos and poor language in some crucial places (I point out some of these instances below). I think this paper would benefit greatly from a more thorough and carefully written technical overview of the algorithm, potentially even in the supplementary material. I would be slightly hesitant to accept the paper in its present state, since I feel a small improvement here would greatly reduce the burden placed on any future reader. Thus, I would consider improving my score with a firm promise to address this issue in the final version. It would have been nice to see some concrete instantiations of Corollary 1, even in the supplementary material. What are the sample complexities for these problems, and how do they compare with Acharya et al.'s work? What type of properties can you handle that they do not (I think distance to uniformity?)? I think it would be worth expanding on the implications of this result. Also, wouldn't entropy depend doubly-logarithmically on the domain size, since f_max can be as large as log k? This is contrary to what is stated in line 173-174. Some typos I discovered (not comprehensive): The crucial definition in line 180 appears to be incorrect at first glance, as the s in the summation is never used. Line 227 typo (deferentially). Line 237 has poor grammar (and generally, this sentence could be expanded a bit -- why does a large L1 distance imply a large sensitivity in the histogram (and what does the sensitivity of a histogram even mean?)). Superscript in line 236: h^2_2 should be h^s_2. Missing period line 243. Algorithm PrivHist has Z ~ G(e^-eps_1) -- should be Z^a. Also, Z^b should have eps_2 instead of eps_1. Input in PrivHist-LowPrivacy H^\ell is missing a c in the superscript. EDIT: Based on the promise of the authors to improve the explanation of the main algorithm, I have raised my score.

Reviewer 2



This paper considers the problem of DP data releasing. Suppose there is some data set, the task is to release the "profile" of it in a DP way. This problem is important in DP and has several applications, e.g., discrete distribution estimation problem. This paper proposes an algorithm which runs in polynomial time and achieves almost optimal l_1 loss, which makes sound theoretical contribution. Their algorithm is based on the following nice observation: there are at most \sqrt{n} different profiles. It loses linear in n (or even worse) by purely releasing profiles or counts, since you have to output all n items in the worst case. But if combining these two in a smarter way, only \sqrt{n} needs to be lost. This paper is well written overall. The writing is generally neat and there are only few typos.

Reviewer 3



The paper is written very clearly, it spends just the right time on giving motivation for the current results, as well as discussing prior work. The derivation of the final algorithm is gradually introduced step-by-step, which for me personally was really important in gaining intuition. As mentioned above, the algorithm closes a gap as its performance in l1 utility matches that of an existing lower bound, which already earned my attention pretty early on. However, I wish there was a bit more motivation for PrivHist, in comparison with the algorithm that achieves a similar rate for (epsilon, delta)-DP with delta>0. I want to confirm - the algorithm you are referring to in “Pure vs approximate differential privacy” has error O(sqrt{n}/epsilon)? In which case, your solution also improves dependence on epsilon? if yes, I think this should be added when discussing prior work. The derivation of PrivHist is non-trivial, and goes beyond mere geometric noise addition. However, given that there has been some prior work on differentially private release of anonymized histograms, I would appreciate if the authors commented on whether these existing algorithms used some of the techniques discussed here - like splitting r around sqrt{n} and using prevalences in one regime and counts in another, or the smoothing idea used to zero out the prevalence vector, etc. I think this discussion would be helpful in assessing the novelty of presented ideas. Another minor question: is there a formal reason why the algorithm splits epsilon across epsilon1, epsilon2 and epsilon3 evenly? To conclude, I think the proposed algorithm is clearly presented, comes with formal guarantees, and moreover the current work has many potential applications as suggested in the paper. What I also really liked is that on several occasions the authors discussed why “natural” solutions don’t work (e.g. why Laplace noise or adding noise to all counts would give a worse rate). For this reason I recommend its acceptance. P.S. Here are a few typos/stylistic comments I had while reading: In Corollary 1: “suppose f-hat satisfies” and “let there exist” On page 6: deferentially private -> differentially private In Definition 1: “if and only if” is somewhat unnecessary since you are using this equality as the definition Post-feedback: The authors addressed my questions and comments in a satisfactory manner, and I think that making the above comparisons with prior work more explicit will only make the paper stronger. I still recommend acceptance.

[Author Response · NeurIPS 2019]

We thank all the reviewers for their comments and valuable feedback which will help improve the paper.

## Reviewer 1

**Explanation of the algorithm**

We thank the reviewer for pointing out the typos. We will definitely improve the writing of the pseudo code and the prose in the final version. If the page limit becomes an issue, we will add a longer exposition in the appendix. We assure that we will address the reviewer's concerns in the final version and ensure that Section 4 and the pseudo code are reader friendly.

**Concrete instances of Corollary 1, comparison to Acharya et al., and other applications**

We will add concrete instantiations of Corollary 1 in the appendix for well studied symmetric functions and compare them to previous works. For a comparison between our work and Acharya et al., observe that since we release the entire histogram, our privacy mechanism can be used for many symmetric properties simultaneously, while Acharya et al.,'s work studies the problem for specific properties. Hence, our result for a specific symmetric property can be slightly worse. For example, consider entropy estimation. The main term in our privacy cost is $\tilde{\mathcal{O}}\left(\left(1/\alpha^2\epsilon\right)^{\frac{1}{1-2\beta}}\right)$ and Acharya et al's bound is $\mathcal{O}\left(1/(\alpha\epsilon)^{1+\beta}\right)$. Thus for $\beta = 0.1$, our dependence on $\epsilon$ and $\alpha$ is slightly worse. We agree with the reviewer that our work should also extend to other properties such distance to uniformity, which to the best of our knowledge has not been studied in the DP framework.

**Doubly logarithmic dependence on $k$ for entropy estimation**

We thank the reviewer for catching this. We agree with the reviewer that dependence on $f_{\max}$ introduces an additive doubly logarithmic dependence on the domain size for entropy. We will modify line 173 to read "Furthermore, the increase is dependent on the maximum value of the function for distributions of interest and it does not explicitly depend on the support size".

## Reviewer 2

**Approximate vs pure DP**

Since pure DP is strictly better than approximate DP, our algorithms also imply approximate DP guarantees. However, previous and our lower bounds do not hold in the approximate DP setting and we plan to pursue this in future. We thank the reviewer for raising this question.

## Reviewer 3

We thank the reviewer for the stylistic comments and typos.

**Comparison to Blocki et al.,'s $(\epsilon, \delta)$-DP result and other approaches**

The algorithm we refer in "Pure vs approximate differential privacy" is due to Block et al., and as the reviewer stated it has an $\ell_1$ error of $\mathcal{O}(\sqrt{n}/\epsilon + \log(1/\delta)/\epsilon)$. We improve on the dependence on $\epsilon$ compared to this work. Furthermore, our $(\epsilon, 0)$-DP guarantee is stronger than the $(\epsilon, \delta)$-DP of Blocki. et al.

We will also discuss previous algorithms and explicitly state which parts of our algorithm are new. To answer the reviewer's question: To the best of our knowledge both (i) splitting $r$ around $\sqrt{n}$ and using prevalences in one regime and counts in another and (ii) the smoothing idea used to zero out the prevalence vector are new and have not been explored before. Of the two (i) is crucial for the computational complexity of the algorithm and (ii) is crucial in improving the $\epsilon$-dependence from $1/\epsilon$ to $1/\sqrt{\epsilon}$ in the high privacy regime ($\epsilon \leq 1$). There are few subtle differences such as cumulative prevalences vs actual prevalences. We will explicitly highlight the above contributions in detail in the final version.

Finally, we note that Blocki et al., proposed an algorithm based on exponential and approximately exponential mechanisms on prevalences, whereas our algorithm is based on Laplace and Geometric mechanisms together with the splitting idea and smoothing methods described above. We will add the above discussion in detail in the paper. We hope that the above discussion clarifies the relation to the Blocki, Datta, and Bonneau paper to our work.

**Even split of $\epsilon$ between $\epsilon_1$, $\epsilon_2$, and $\epsilon_3$**

There is no particular reason for $\epsilon_1$, $\epsilon_2$, and $\epsilon_3$ to be equal and we chose those values for simplicity and easy readability. We will add a discussion in the appendix on better ways of splitting the privacy budget. For example, since $\epsilon_1$ is just used to estimate $n$, analysis of the algorithm shows that $\epsilon_2, \epsilon_3$ affects utility more than $\epsilon_1$. Hence, we can set $\epsilon_2 = \epsilon_3 = \epsilon(1 - o(1))/2$ and $\epsilon_1 = o(\epsilon)$ to get better practical results.

[Meta-Review · NeurIPS 2019]

All reviewers consider the paper strong and recommend acceptance.